# CHIP: A Hawkes Process Model for Continuous-time Networks with Scalable and Consistent Estimation

**Makan Arastuie**
EECS Department
University of Toledo
makan.arastuie
@rockets.utoledo.edu

**Subhadeep Paul**
Department of Statistics
The Ohio State University
paul.963@osu.edu

**Kevin S. Xu**
EECS Department
University of Toledo
kevin.xu@utoledo.edu

## Abstract

In many application settings involving networks, such as messages between users of an on-line social network or transactions between traders in financial markets, the observed data consist of timestamped relational events, which form a continuous-time network. We propose the *Community Hawkes Independent Pairs (CHIP)* generative model for such networks. We show that applying spectral clustering to an aggregated adjacency matrix constructed from the CHIP model provides *consistent community detection* for a growing number of nodes and time duration. We also develop consistent and computationally efficient estimators for the model parameters. We demonstrate that our proposed CHIP model and estimation procedure scales to large networks with tens of thousands of nodes and provides superior fits than existing continuous-time network models on several real networks.

## 1 Introduction

A variety of complex systems in the computer, information, biological, and social sciences can be represented as a network, which consists of a set of objects (nodes) and relationships (edges) between the objects. In many application settings, we observe edges in the form of distinct events occurring between nodes over time. For example, in on-line social networks, users interact with each other through events that occur at specific time instances such as liking, mentioning, or sharing another user's content. Such interactions form *timestamped relational events*, where each event is a triplet $(i, j, t)$ denoting events from node $i$ (sender) to node $j$ (receiver) at timestamp $t$. The observation of these triplets defines a dynamic network that continuously evolves over time.

Timestamped relational event data are usually modeled by combining a point process model for the event times with a network model for the sender and receiver [1–9]. We refer to such models as *continuous-time network models* because they provide probabilities of observing events between two nodes during arbitrarily short time intervals. For model-based exploratory analysis and prediction of future events with relational event data, continuous-time network models are often superior to their discrete-time counterparts [10–14], which first aggregate events over time windows to form discrete-time network "snapshots" and thus lose granularity in modeling temporal dynamics.

We propose the *Community Hawkes Independent Pairs (CHIP)* model, which is inspired by the recently proposed Block Hawkes Model (BHM) [9] for timestamped relational event data. Both CHIP and BHM are based on the Stochastic Block Model (SBM) for static networks [15]. In the BHM, events between different pairs of nodes belonging to the same pair of communities are dependent, which makes it difficult to analyze. In contrast, for CHIP the pairs of nodes in the same community generate events according to *independent* univariate Hawkes processes with shared parameters, so that the number of parameters remains the same as in the BHM. The independence between node pairs enables tractable analysis of the CHIP model and more scalable estimation than the BHM.

Our main contributions are as follows. (1) We demonstrate that spectral clustering provides consistent community detection in the CHIP model for a growing number of nodes and time duration. (2) We propose consistent and computationally efficient estimators for the model parameters also for a growing number of nodes and time duration. (3) We show that the CHIP model provides better fits to several real datasets and scales to much larger networks than existing models, including a Facebook network with over $40,000$ nodes and over $800,000$ events. Other point process network models have demonstrated good empirical results, but to the best of our knowledge, this work provides the *first theoretical guarantee of estimation accuracy*. Our asymptotic analysis also has tremendous practical value given the scalability of our model to large networks with tens of thousands of nodes.

## 2 Background

### 2.1 Hawkes Processes

The Hawkes process [16] is a counting process designed to model continuous-time arrivals of events that naturally cluster together in time, where the arrival of an event increases the chance of the next event arrival immediately after. They have been used to model earthquakes [17], financial markets [18, 19], and user interactions on social media [3, 20].

A univariate Hawkes process is a *self-exciting* point process where its conditional intensity function given a sequence of event arrival times $\{t_1, t_2, t_3, ..., t_l\}$ for $l$ events up to time duration $T$ takes the general form $\lambda(t) = \mu + \sum_{t_i < t}^{t_l} \gamma(t - t_i)$, where $\mu$ is the background intensity and $\gamma(\cdot)$ is the kernel or the excitation function. A frequent choice of kernel is an exponential kernel, parameterized by $\alpha, \beta > 0$ as $\gamma(t - t_i) = \alpha e^{-\beta(t - t_i)}$, where the arrival of an event instantaneously increases the conditional intensity by the jump size $\alpha$, after which the intensity decays exponentially back towards $\mu$ at rate $\beta$. Restricting $\alpha < \beta$ yields a stationary process. We use an exponential kernel for the CHIP model, since it has been shown to provide a good fit for relational events in social media [9, 21–23].

### 2.2 The Stochastic Block Model

Statistical models for networks typically consider a static network rather than a network of relational events. Many static network models are discussed in the survey by Goldenberg et al. [24]. A static network with $n$ nodes can be represented by an $n \times n$ adjacency matrix $A$ where $A_{ij} = 1$ if there is an edge between nodes $i$ and $j$ and $A_{ij} = 0$ otherwise. We consider networks with no self-edges, so $A_{ii} = 0$ for all $i$. For a directed network, we let $A_{ij} = 1$ if there is an edge from node $i$ to node $j$.

One model that has received significant attention is the *stochastic block model* (SBM), formalized by Holland et al. [15]. In the SBM, every node $i$ is assigned to one and only one community or *block* $c_i \in \{1, \ldots, k\}$, where $k$ denotes the total number of blocks. For a directed SBM, given the block membership vector $\mathbf{c} = [c_i]_{i=1}^n$, all off-diagonal entries of the adjacency matrix $A_{ij}$ are independent Bernoulli random variables with parameter $p_{c_i, c_j}$, where $p$ is a $k \times k$ matrix of probabilities. Thus the probability of forming an edge between nodes $i$ and $j$ depends only on the block memberships $c_i$ and $c_j$. There have been significant advancements in the analysis of estimators for the SBM. Several variants of spectral clustering [25], including regularized versions [26, 27], have been shown to be consistent estimators of the community assignments in the SBM and various extensions in several asymptotic settings [28–39]. Spectral clustering scales to large networks with tens of thousands of nodes and is generally not sensitive to initialization, so it is also a practically useful estimator.

### 2.3 Related Work

One approach for modeling continuous-time networks is to treat the edge strength of each node pair as a continuous-time function that increases when an event occurs between the node pair and then decays afterwards [40–42]. Another approach is to combine a point process model for the event times, typically some type of Hawkes process, with a network model. The conditional intensity functions of the point processes then serve as the time-varying edge strengths. Point process network models are used in two main settings. The first involves estimating the structure of a latent or unobserved network from observed events at the nodes [43–48]. These models are often used to estimate *static* networks of diffusion from information cascades.

In the second setting, which we consider in this paper, we directly observe events *between pairs of nodes* so that events take on the form $(i, j, t)$ denoting an event from node $i$ to node $j$ at timestamp $t$. Our objective is to model the dynamics of such event sequences. In many applications, including messages on on-line social networks, most pairs of nodes either never interact and thus have no events between them. Thus, most prior work in this setting utilizes low-dimensional latent variable representations of the networks to parameterize the point processes.

The latent variable representations are often inspired by generative models for static networks such as continuous latent space models [49] and stochastic block models [15], resulting in the development of point process network models with continuous latent space representations [6] and latent block or community representations [1–5, 7–9], respectively. Point process network models with latent community representations are most closely related to the model we consider in this paper. Exact inference in such models is intractable due to the discrete nature of the community assignments. Approximate inference techniques including Markov Chain Monte Carlo (MCMC) [2, 3, 7] or variational inference [5, 8, 9] have been used in prior work. While such techniques have demonstrated good empirical results, to the best of our knowledge, they come with no theoretical guarantees.

## 3   The Community Hawkes Independent Pairs (CHIP) Model

We consider a generative model for timestamped relational event networks that we call the *Community Hawkes Independent Pairs (CHIP)* model. The CHIP model has parameters $(\boldsymbol{\pi}, \mu, \alpha, \beta)$. Each node is assigned to a community or block $a \in \{1, \ldots, k\}$ with probability $\pi_a$, where each entry of $\boldsymbol{\pi}$ is non-negative and all entries sum to 1. We represent the block assignments of all nodes either by a length $n$ vector $\mathbf{c} = [c_i]_{i=1}^n$ or an $n \times k$ binary matrix $C$ where $c_i = q$ is equivalent to $C_{iq} = 1$, $C_{iq'} = 0$ for all $q' \neq q$. Each of the parameters $\mu, \alpha, \beta$ is a $k \times k$ matrix. While we assume that the number of blocks and the block assignments of the nodes do not change with time, the CHIP model captures time-varying behavior due to the incorporation of self-exciting point processes. Event times between node pairs $(i, j)$ within a block pair $(a, b)$ follow independent exponential Hawkes processes with shared parameters: baseline rate $\mu_{ab}$, jump size $\alpha_{ab}$, and decay rate $\beta_{ab}$. The generative process for our proposed CHIP model is as follows:

$$
\begin{aligned}
c_i &\sim \text{Categorical}(\boldsymbol{\pi}) && \text{for all nodes } i \\
\mathbf{t}_{ij} &\sim \text{Hawkes process}(\mu_{c_i c_j}, \alpha_{c_i c_j}, \beta_{c_i c_j}) && \text{for all } i \neq j \\
Y &= \text{Row concatenate } [(i\mathbf{1}, j\mathbf{1}, \mathbf{t}_{ij})] && \text{over all } i \neq j
\end{aligned}
$$

$\mathbf{1}$ denotes the all-ones vector of appropriate length. Let $T$ denote the end time of the Hawkes process, which would correspond to the duration of the data trace. The column vector of event times $\mathbf{t}_{ij}$ has length $N_{ij}(T)$, which denotes the number of events from node $i$ to node $j$ up to time $T$. Let $Y$ denote the event matrix with dimension $l \times 3$, where $l = \sum_{i,j} N_{ij}(T)$ denotes the total number of observed events over all node pairs. It is constructed by row concatenating triplets $(i, j, t_{ij}(q))$ over all events $q \in \{1, \ldots, N_{ij}(T)\}$ for all node pairs $i, j \in \{1, \ldots, n\}, i \neq j$.

### 3.1   Relation to Other Models

Our proposed CHIP model has a generative structure inspired by the SBM for static networks. Other point process network models in the literature have also utilized similar block structures, but they have been incorporated in two different approaches. One approach involves placing point process models at the level of block pairs [2, 4, 5, 9]. For a network with $k$ blocks, $k^2$ different point processes are used to generate events between the $k^2$ block pairs. To generate events between pairs of nodes, rather than pairs of blocks, the point processes are thinned by randomly selecting nodes from the respective blocks so that all nodes in a block are stochastically equivalent, in the spirit of the SBM. Such models have demonstrated good empirical results, but the dependency between node pairs complicates analysis of the models.

The other approach involves modeling pairs of nodes with independent point processes that share parameters among nodes in the same block [1, 3, 8]. By having node pairs in the same block share parameters, the number of parameters is the same as for the models with block pair-level point processes. However, by using independent point processes for all node pairs, there is no dependency between node pairs, which simplifies analysis of the model. We use this approach in the proposed CHIP model and exploit this independence to perform the theoretical analysis in Section 4.

---
**Algorithm 1** Estimation procedure for Community Hawkes Independent Pairs (CHIP) model
---
**Input:** Relational event matrix $Y$, number of blocks $k$

**Result:** Estimated block assignments $\hat{C}$ and CHIP model parameters $(\hat{\boldsymbol{\pi}}, \hat{\mu}, \hat{\alpha}, \hat{\beta})$

1: **for all** node pairs $i \neq j$ **do**
2:     $N_{ij}$ = number of events from $i$ to $j$ in $Y$
3: $\hat{C} \leftarrow$ Spectral clustering$(N, k)$
4: **for all** block pairs $(a, b)$ **do**
5:     Compute estimates $(\hat{m}_{ab}, \hat{\mu}_{ab})$ using (2)
6:     $\hat{\beta}_{ab} \leftarrow$ maximize log-likelihood by line search
7:     $\hat{\alpha}_{ab} \leftarrow \hat{\beta}_{ab} \hat{m}_{ab}$
8: $\hat{\boldsymbol{\pi}} \leftarrow$ proportion of nodes in each block
9: **return** $[\hat{C}, \hat{\boldsymbol{\pi}}, \hat{\mu}, \hat{\alpha}, \hat{\beta}]$
---

### 3.2 Estimation Procedure

As with many other block models, the maximum-likelihood estimator for the discrete community assignments $C$ is intractable except for extremely small networks (e.g. 20 nodes). We propose a scalable estimation procedure for the CHIP model that has two components as shown in Algorithm 1: community detection and parameter estimation. For the community detection component, we use spectral clustering on the weighted adjacency or count matrix $N(T)$ or simply $N$ with entries $N_{ij}(T)$. Since this is a directed adjacency matrix, we use singular vectors rather than eigenvectors for spectral clustering (see Algorithm A.1 in the supplementary material for details).

For the parameter estimation component, we first consider estimating the Hawkes process parameters $(\mu_{ab}, \alpha_{ab}, \beta_{ab})$ for each block pair $(a, b)$ using only the count matrix $N$, which discards event timestamps. Even without the event timestamps, we are able to estimate $\mu_{ab}$ and the ratio $m_{ab} = \alpha_{ab}/\beta_{ab}$, but not the parameters $\alpha_{ab}$ and $\beta_{ab}$ separately. Define the following terms, which are the sample mean and (unbiased) sample variance of the pairwise event counts within each block pair:

$$\bar{N}_{ab} = \frac{1}{n_{ab}} \sum_{i,j:C_{ia}=1, C_{jb}=1} N_{ij}, \quad S_{ab}^2 = \frac{1}{n_{ab}-1} \sum_{i,j:C_{ia}=1, C_{jb}=1} (N_{ij} - \bar{N}_{ab})^2, \tag{1}$$

where $n_{ab}$ denotes the number of node pairs in block pair $(a, b)$ and is given by $n_{ab} = |a||b|$ for $a \neq b$ and $n_{ab} = |a||a-1|$ for $a = b$, with $|a|$ denoting the number of nodes in block $a$. $\bar{N}_{ab}$ and $S_{ab}^2$ are unbiased estimators of the mean and variance, respectively, of the counts of the number of events between all node pairs $(i, j)$ in block pair $(a, b)$. Using $\bar{N}_{ab}$ and $S_{ab}^2$, we propose the following method of moments estimators (conditioned on the estimated blocks) for $m_{ab}$ and $\mu_{ab}$ from the count matrix $N$:

$$\hat{m}_{ab} = 1 - \sqrt{\frac{\bar{N}_{ab}}{S_{ab}^2}}, \quad \hat{\mu}_{ab} = \frac{1}{T} \sqrt{\frac{(\bar{N}_{ab})^3}{S_{ab}^2}}. \tag{2}$$

Finally, the vector of block assignment probabilities $\boldsymbol{\pi}$ can be easily estimated using the proportion of nodes in each block, i.e. $\hat{\pi}_a = \frac{1}{n} \sum_{i=1}^{n} \hat{C}_{ia}$ for all $a = 1, \ldots, k$.

In some prior work, exponential Hawkes processes are parameterized only in terms of $m$ and $\mu$, with $\beta$ treated as a known parameter that is not estimated [50–52]. In this case, the estimation procedure is complete. On the other hand, if we want to estimate the values of both $\alpha$ and $\beta$ rather than just their ratio, we have to use the actual event matrix $Y$ with the event timestamps. To separately estimate the $\alpha_{ab}$ and $\beta_{ab}$ parameters, we replace $\alpha_{ab} = \beta_{ab} m_{ab}$ in the exponential Hawkes log-likelihood for block pair $(a, b)$ then plug in our estimate $\hat{m}_{ab}$ for $m_{ab}$. Then the log-likelihood is purely a function of $\beta_{ab}$ and can be maximized using a standard scalar optimization or line search method.

### 3.3 Selection of the Number of Blocks

The estimation procedure in Algorithm 1 assumes that the number of blocks $k$ is provided. In many practical settings, $k$ is unknown, and choosing $k$ becomes a model selection problem. Given that CHIP uses spectral clustering on the weighted adjacency matrix $N$, model selection approaches for static block models can be used to find the optimal $k$. These range in complexity from the eigengap

heuristic [25] to more sophisticated methods including using eigenvalues of the non-backtracking matrix and Bethe Hessian matrix [53] and network cross validation [54, 55]. Another approach, specific to the timestamped network setting we consider in this paper, is to hold out a portion of the events, e.g. the last 20%, and select the $k$ that maximizes the log-likelihood on the held-out events.

## 4 Theoretical Analysis of Estimators

### 4.1 Analysis of Estimated Community Assignments

We define the error of community detection as the misclustering error rate $r = \inf_\Pi \frac{1}{n} \sum_{i=1}^n 1(c_i \neq \Pi(\hat{c}_i))$, where $\Pi(\cdot)$ denotes the set of all permutations of the community labels. Our proposed CHIP model considers directed events; however, we analyze community detection on undirected networks to better match up with the literature on analysis of spectral clustering for the SBM. The bounds and consistency properties we derive still apply to the directed case with only a change in the constants. We assume that $T \to \infty$, which can be achieved by rescaling the time unit for event times. Under this assumption, the mean and variance of the number of events between nodes $(i, j)$ are [56–58]

$$\nu_{ab} = \frac{\mu_{ab}T}{1 - \alpha_{ab}/\beta_{ab}}, \quad \sigma_{ab}^2 = \frac{\mu_{ab}T}{(1 - \alpha_{ab}/\beta_{ab})^3}. \tag{3}$$

We analyze community detection error in a simplified special case of our CHIP model which is in similar spirit to a commonly-employed case in the stochastic block models literature [28, 31, 32, 35, 59]. We provide analogous results for the general CHIP model in Section B.1 of the supplementary material. In this special case, all communities have roughly equal number of elements $|a| \asymp n/k$, all intra-community processes (diagonal block pairs) have the same set of parameters $\mu_1, \alpha_1, \beta_1$ and all inter-community processes (off-diagonal block pairs) have the same set of parameters $\mu_2, \alpha_2, \beta_2$. We use the notation $Y \sim \text{CHIP}(C, n, k, \mu_1, \alpha_1, \beta_1, \mu_2, \alpha_2, \beta_2)$ to denote a relational event matrix $Y$ generated from this simplified model. Define $m_1 = \alpha_1/\beta_1$ and $m_2 = \alpha_2/\beta_2$. Let $\nu_1 = \mu_1/(1 - m_1)$ and $\nu_2 = \mu_2/(1 - m_2)$, while $\sigma_1^2 = \mu_1/(1 - m_1)^3$ and $\sigma_2^2 = \mu_2/(1 - m_2)^3$. Assume $\nu_1 > \nu_2$, $\nu_1 \asymp \nu_2$, and $\sigma_1 \asymp \sigma_2$, where the asymptotic equivalence is with respect to both $n$ and $T$. These assumptions imply that the expected number of events are higher between two nodes in the same community compared to two nodes in different communities and that the asymptotic dependence on $n$ and $T$ are the same for both set of parameters. This setting is useful to understand detectability limits and has been widely employed in the literature on stochastic block models [32, 35, 36, 59–61]. In this setting, we have the following upper bound on the misclustering error rate.

**Theorem 1** *Let $Y \sim \text{CHIP}(C, n, k, \mu_1, \alpha_1, \beta_1, \mu_2, \alpha_2, \beta_2)$. The misclustering error rate for spectral clustering on the weighted adjacency matrix $N$ at time $T \to \infty$ is*

$$r \lesssim \frac{T\sigma_1^2 n}{(n/k)^2 (\nu_2 - \nu_1)^2 T^2} \asymp \frac{k^2}{nT} \frac{\sigma_1^2}{(\nu_1 - \nu_2)^2}.$$

We note that if the set of parameters $\mu, \alpha, \beta$ remain constant as a function of $n$ and $T$ then the misclustering error rate decreases as $1/T$ with increasing $T$, decreases as $1/n$ with increasing $n$, and increases as $k^2$ with increasing $k$. Hence, as we observe the process for more time, spectral clustering on $N$ has lower error rate. The rate of convergence with increasing $T$ is the same as one would obtain for detecting an average community structure if discrete snapshots of the network were available over time [38, 39, 59]. The dependence of the misclustering error rate on $n$ and $k$ is what one would expect from the SBM literature.

Theorem 1 applies also in the sparse graph setting. We let $\mu \asymp 1/[f(n)g(T)]$, a function of $n$ and $T$, and explore various sparsity settings by varying $f$ and $g$ in Section B.1.1 of the supplementary material. Our proofs allow $\mu$ to vary with $n$ and $T$ and can be as small as $\log(n)/(nT)$. A key component in the proof of Theorem 1 is a bound from Bandeira and van Handel [62]. In Section B.1 of the supplementary material, we provide the proof of Theorem 1, an analogous theorem for the general CHIP model, as well as theorems for spectral clustering on an unweighted adjacency matrix.

### 4.2 Analysis of Estimated Hawkes Process Parameters

As discussed in Section 3.2, we are able to estimate $m = \alpha/\beta$ and $\mu$ from the count matrix $N$ using (2). We analyze these estimators assuming a growing number of nodes $n$ and time duration

$T$. We do not put any assumption on the distribution of the counts; we only require that $T$ is large enough such that the asymptotic mean and variance equations in (3) hold. The sample mean $\bar{N}_{ab}$ and sample variance $S_{ab}^2$ of the counts are unbiased estimators of $\nu_{ab}$ and $\sigma_{ab}^2$, respectively. The following theorem shows that these estimators are consistent and asymptotically normal.

**Theorem 2** *Define $n_{\min} = \min_{a,b} n_{ab}$. The estimators for $m_{ab}$ and $\mu_{ab}$ have the following asymptotic distributions as $n_{\min} \to \infty$ and $T \to \infty$:*

$$\sqrt{n_{ab}}\left(\hat{m}_{ab} - \left(1 - \sqrt{\frac{\nu_{ab}}{\sigma_{ab}^2}}\right)\right) \xrightarrow{d} \mathcal{N}\left(0, \frac{1}{4\nu_{ab}}\right), \ \sqrt{n_{ab}}\left(\hat{\mu}_{ab}T - \frac{(\nu_{ab})^{3/2}}{\sigma_{ab}}\right) \xrightarrow{d} \mathcal{N}\left(0, \frac{9}{4}\nu_{ab}\right).$$

Using Theorem 2, we obtain confidence intervals for $\mu$ and $m$, in Section B.2.1 of the supplementary material. In the simplified special case of Theorem 1, we have equal community sizes so $n_{ab} \asymp (n/k)^2$. Therefore, the condition $n_{\min} \to \infty$ boils down to $(n/k)^2 \to \infty$, which is a reasonable assumption. Theorem 2 guarantees convergence of our estimators for $\mu$ and $m$ with the asymptotic mean-squared errors (MSEs) decreasing at the rate $n_{ab} \asymp (n/k)^2$ under the assumption that the community structure is correctly estimated. Next, we provide an "end-to-end" guarantee for the convergence of the asymptotic MSE to 0 for estimating the mean number of events in each block pair $\nu_{ab}$ using the sample mean $\bar{N}_{ab}$ incorporating the error in estimating communities using spectral clustering from Theorem 1.

**Theorem 3** *Assume $n_{ab} \asymp (n/k)^2$. The weighted average of asymptotic MSEs in estimating $\nu_{ab}$ using the estimator $\bar{N}_{ab}$ with communities estimated by spectral clustering is*

$$\frac{\sum_{ab} n_{ab} E[(\bar{N}_{ab} - \nu_{ab})^2]}{\sum_{ab} n_{ab}} \lesssim \frac{kT}{n} \max\left\{\sigma_1^2, \frac{k^2\sigma_1^2\nu_2^2}{(\nu_1 - \nu_2)^2}\right\}.$$

*For comparison, under the assumption that the community structure is correctly estimated, the weighted average of asymptotic MSEs in estimating $\nu_{ab}$ using the estimator $\bar{N}_{ab}$ is*

$$\frac{\sum_{ab} n_{ab} E[(\bar{N}_{ab} - \nu_{ab})^2]}{\sum_{ab} n_{ab}} = \frac{k^2 T \sigma_1^2}{n^2}.$$

Theorem 3 guarantees that the MSE for estimating Hawkes process parameters decreases at least at a linear rate with increasing $(n/k)$ when the error from community detection is taken into account instead of the quadratic rate when the error is not taken into account. The proofs for Theorems 2 and 3 are provided in Section B.2.2 of the supplementary material.

## 5 Experiments

We begin with a set of simulation experiments to assess the accuracy of our proposed estimation procedure and verify our theoretical analysis. We then present several experiments on real data involving both prediction and model-based exploratory analysis. Additional experiments are provided in Section C of the supplementary material, along with the code[1] to replicate all experiments.

### 5.1 Community Detection on Simulated Networks with Varying $T$, $n$, and $k$

We simulate networks from the simplified CHIP model while varying two of $T$, $n$, and $k$ simultaneously. We choose parameters $\mu_1 = 0.085$, $\mu_2 = 0.065$, $\alpha_1 = \alpha_2 = 0.06$, and $\beta_1 = \beta_2 = 0.08$. The upper bounds on the error rates in Theorem 1 involve all three parameters $n, k, T$ simultaneously, making it difficult to interpret the result. To better observe the effects of $n, k, T$ and their relationship with respect to each other, we perform three separate simulations each time varying two and fixing the other one. The community detection accuracy averaged over 30 simulations using the weighted adjacency matrix $N$ as two of $T$, $n$, and $k$ are varied is shown in Figure 1. Since the estimated community assignments will be permuted compared to the actual community labels, we evaluate the community detection accuracy using the adjusted Rand score [63], which is 1 for perfect community detection and has an expectation of 0 for a random assignment.

Note that Theorem 1 predicts that the misclustering error rate varies as $k^2/(nT)$ if all three parameters are varied. Figure 1(a) shows the accuracy to be low for small $T$ and large $k$. As we simultaneously

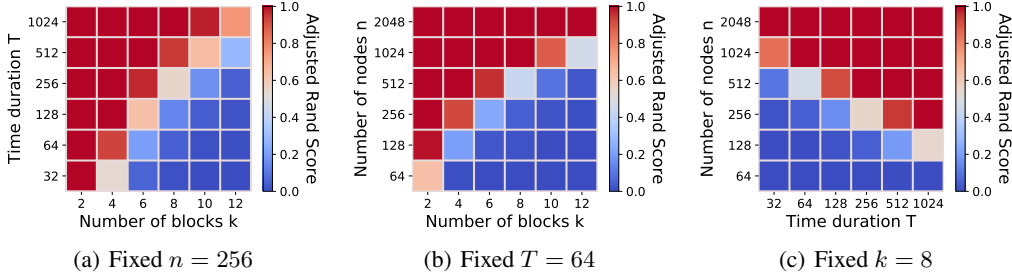

(a) Fixed $n = 256$　　　　　(b) Fixed $T = 64$　　　　　(c) Fixed $k = 8$

Figure 1: Heat map of adjusted Rand score of spectral clustering on weighted adjacency matrix, with varying $T$, $n$, and $k$, averaged over 30 simulated networks.

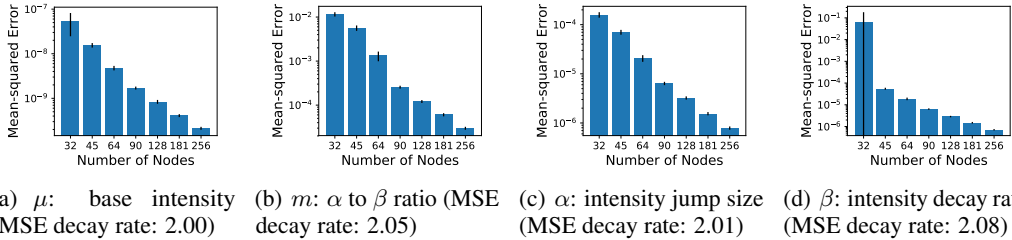

(a) $\mu$: base intensity (MSE decay rate: 2.00)　(b) $m$: $\alpha$ to $\beta$ ratio (MSE decay rate: 2.05)　(c) $\alpha$: intensity jump size (MSE decay rate: 2.01)　(d) $\beta$: intensity decay rate (MSE decay rate: 2.08)

Figure 2: Mean-squared errors (MSEs) of CHIP's Hawkes parameter estimators averaged over 100 simulations ($\pm$ 2 standard errors) on a log-log plot. MSEs for all four parameters decreases as the number of nodes increases, with the estimated decay rate (exponent) beginning at 90 nodes listed.

increase $T$ and decrease $k$ the accuracy improves until the adjusted Rand score reaches 1. We also note that it is possible to obtain high accuracy either with increasing $T$ or decreasing $k$ or with both even when $n$ is fixed. This is in line with the prediction from Theorem 1 that the misclustering error rate varies as $k^2/T$ if $n$ remains fixed. We observe a similar effect of increasing accuracy with increasing $n$ and decreasing $k$ when $T$ is kept fixed in Figure 1(b). Finally, Figure 1(c) verifies the prediction that accuracy increases with both increasing $n$ and $T$ for a fixed $k$.

## 5.2　Hawkes Process Parameter Estimation on Simulated Networks

Next, we examine the estimation accuracy of the Hawkes process parameter estimates as described in Section 4.2. We simulate networks from the simplified CHIP model with $k = 4$ blocks, duration $T = 10,000$ and parameters $\mu_1 = 0.0011$, $\mu_2 = 0.0010$, $\alpha_1 = 0.11$, $\alpha_2 = 0.09$, $\beta_1 = 0.14$, and $\beta_2 = 0.16$ so that each parameter is different between block pairs. We then run the CHIP estimation procedure: spectral clustering followed by Hawkes process parameter estimation.

Figure 2 shows the mean-squared errors (MSEs) of all four estimators decay quadratically as $n$ increases. Theorem 2 states that $\hat{m}$ and $\hat{\mu}$ are consistent estimators with MSE decreasing at a quadratic rate for growing $n$ with known communities. Here, we observe the quadratic decay even with communities estimated by spectral clustering, where the mean adjusted Rand score is increasing from 0.6 to 1 as $n$ grows. We observe that $\alpha$ and $\beta$ are also accurately estimated for growing $n$ even though $\beta$ is estimated using a line search for which we have no guarantees.

## 5.3　Comparison with Other Models on Real Networks

We perform experiments on three real network datasets. Each dataset consists of a set of events where each event is denoted by a sender, a receiver, and a timestamp. The MIT Reality Mining [64] and Enron [65] datasets were loaded and preprocessed identically to DuBois et al. [3] to allow for a fair comparison with their reported values. On the Facebook wall posts dataset [66], we use the largest connected component of the network excluding self loops ($43,953$ nodes). Additional details about the datasets and preprocessing are provided in Section C.2.1 of the supplementary material.

Table 1: Mean test log-likelihood per event for each real network dataset across all models. Larger (less negative) values indicate better predictive ability. Bold entry denotes best fit for a dataset. Results for REM are reported values from DuBois et al. [3]. Poisson denotes spectral clustering followed by estimating a Poisson process baseline model. *The BHM local search does not scale up to the Facebook network, so we report results using the (less accurate) spectral clustering procedure.

| Dataset | Statistics | Model | $k = 1$ | $k = 2$ | $k = 3$ | $k = 10$ | Best $k$ |
|---------|-----------|-------|---------|---------|---------|----------|----------|
| Reality | $n = 70$ $l_{\text{train}} = 1{,}500$ $l_{\text{test}} = 661$ | CHIP | $-4.83$ | $-4.88$ | $-5.06$ | $-6.69$ | $-\mathbf{4.83}$ ($k = 1$) |
| | | REM | $-6.78$ | $-7.42$ | $-6.11$ | $-6.61$ | $-6.11$ ($k = 3$) |
| | | BHM | $-9.05$ | $-7.56$ | $-7.60$ | $-5.74$ | $-5.37$ ($k = 50$) |
| | | Poisson | $-10.3$ | $-10.4$ | $-9.63$ | $-9.38$ | $-8.51$ ($k = 32$) |
| Enron | $n = 142$ $l_{\text{train}} = 3{,}000$ $l_{\text{test}} = 1{,}000$ | CHIP | $-5.63$ | $-5.61$ | $-5.65$ | $-7.15$ | $-\mathbf{5.61}$ ($k = 2$) |
| | | REM | $-7.02$ | $-6.86$ | $-6.84$ | $-7.26$ | $-6.84$ ($k = 3$) |
| | | BHM | $-8.72$ | $-8.43$ | $-8.39$ | $-7.93$ | $-7.49$ ($k = 8$) |
| | | Poisson | $-11.9$ | $-11.4$ | $-11.5$ | $-12.0$ | $-11.4$ ($k = 4$) |
| Facebook | $n = 43{,}953$ $l_{\text{train}} = 682{,}266$ $l_{\text{test}} = 170{,}567$ | CHIP | $-9.54$ | $-9.58$ | $-9.58$ | $-9.61$ | $-\mathbf{9.46}$ ($k = 9$) |
| | | BHM* | $-16.0$ | $-15.7$ | $-16.2$ | $-14.7$ | $-14.4$ ($k = 22$) |
| | | Poisson | $-20.8$ | $-21.1$ | $-21.1$ | $-20.6$ | $-19.2$ ($k = 55$) |

We fit our proposed Community Hawkes Independent Pairs (CHIP) model as well as the Block Hawkes Model (BHM) [9] to all three real datasets and evaluate their fit. We also compare against a simpler baseline: spectral clustering with a homogeneous Poisson process for each node pair. For each model, we also compare against the case $k = 1$, where no community detection is being performed. We do not have ground truth community labels for these real datasets so we cannot evaluate community detection accuracy. Instead, we use the mean test log-likelihood per event as the evaluation metric, which allows us to compare against the reported results in DuBois et al. [3] for the relational event model (REM). Since the log-likelihood is computed on the test data, this is a measure of the model's ability to *forecast future events* rather than detect communities.

As shown in Table 1, CHIP outperforms all other models in all three datasets. Note that test log-likelihood is maximized for CHIP at relatively small values of $k$ compared to the BHM. This is because CHIP assumes independent node pairs whereas the BHM assumes all node pairs in a block pair are dependent. Thus, the BHM needs a higher value for $k$ in order to model independence. This difference is particularly visible for the Reality Mining data, where CHIP with $k = 1$ is the best predictor of the test data, while the best BHM has $k = 50$ on a network with only 70 nodes! These both suggest a weak community structure that is not predictive of future events in the Reality Mining data, whereas community structure does appear to be predictive in the Enron and Facebook data.

In addition to the improved predictive ability of CHIP compared to the BHM, the computational demand is also significantly decreased. Fitting the CHIP model for each value of $k$ took on average 0.15 s and 0.3 s on the Reality Mining and Enron datasets, respectively, while the BHM took on average 250 s and 30 m, mostly due to the time-consuming local search[2]. We did not implement the MCMC-based inference procedure for the REM and thus do not have results for REM on the Facebook data or computation times. The approach of holding out a test set of events and evaluating test log-likelihood can also be used for selection of the number of blocks $k$. As shown in Figure 3, on the Facebook data, there is hardly any increase in the runtime of CHIP for $k < 100$, and it is manageable even for $k = 1{,}000$.

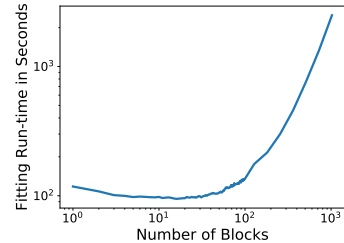

Figure 3: CHIP's fitting runtime on the Facebook data on a log-log scale with increasing $k$.

## 5.4 Model-Based Exploratory Analysis of Facebook Wall Post Network

We use CHIP to perform model-based exploratory analysis to understand the behavior of different groups of users in the Facebook wall post network. We consider all 852,833 events and choose $k = 10$ blocks using the eigengap heuristic [25], which required 141 s to fit. Note that the CHIP

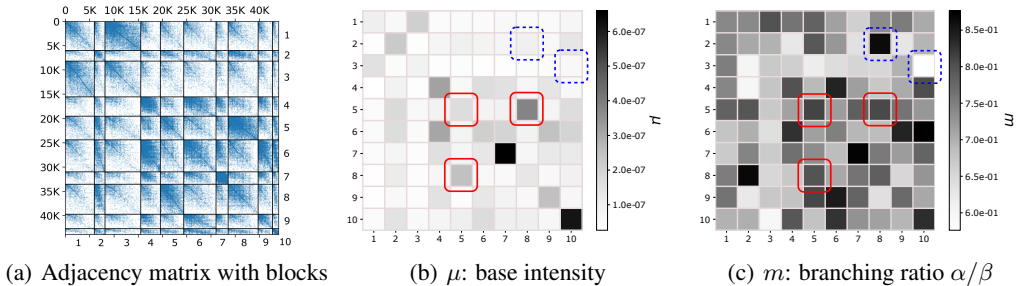

(a) Adjacency matrix with blocks      (b) $\mu$: base intensity      (c) $m$: branching ratio $\alpha/\beta$

Figure 4: Inferred CHIP parameters on the largest connected component of the Facebook Wall Posts dataset with $k = 10$. Axis labels denote block numbers. Each tile corresponds to a block pair where $(a, b)$ denotes row $a$ and column $b$. Boxed block pairs in the heatmap are discussed in the body text.

estimation procedure can scale up to a much higher number of communities also—fitting CHIP to the Facebook data with $k = 1,000$ communities took just under 50 minutes! The adjacency matrix permuted by the block structure is shown in Figure 4(a), and heatmaps of the fitted CHIP parameters are shown in Figures 4(b) and 4(c). Diagonal block pairs on average have a base intensity $\mu$ of $2.8 \times 10^{-7}$, which is higher compared to $9.5 \times 10^{-8}$ for off-diagonal block pairs, indicating an underlying assortative community structure. However, not all blocks have higher rates of within-block posts, e.g. $\mu_{5,8} > \mu_{5,5}$ and $\mu_{8,5} > \mu_{5,5}$, as shown in red boxes in Figure 4(b), which illustrates that the CHIP model does not discourage inter-block events. These patterns often occur in social networks, for instance, if there are communities with opposite views on a particular subject.

While the structure of $\mu$ reveals insights on the baseline rates of events between block pairs, the structure of the branching ratio $m = \alpha/\beta$ shown in Figure 4(c) reveals insights on the burstiness of events between block pairs. For some block pairs, such as $(3, 10)$, there are very low values of $\alpha$ and $\beta$ indicating the events are closely approximated by a homogeneous Poisson process, while some block pairs such as $(2, 8)$ are extremely bursty despite low baseline rates. Both block pairs are shown in blue dashed boxes. The different levels of burstiness of block pairs cannot be seen from aggregate statistics such as the the count matrix $N$.

# 6 Conclusion

We introduced the Community Hawkes Independent Pairs (CHIP) model for timestamped relational event data. The CHIP model has many similarities with the Block Hawkes Model (BHM) [9]; however, in the CHIP model, events among any two node pairs are independent, which enables both tractable theoretical analysis and scalable estimation. We demonstrated that an estimation procedure using spectral clustering followed by Hawkes process parameter estimation provides consistent estimates of the communities and Hawkes process parameters for a growing number of nodes and time duration. Lastly, we showed that CHIP also provides better fits to several real networks compared to the Relational Event Model [3] and the BHM. It also scales to considerably larger data sets, including a Facebook wall post network with over $40,000$ nodes and $800,000$ events.

There are several limitations to the CHIP model and our proposed estimation procedure. Assuming all node pairs to have independent Hawkes processes simplifies analysis and increases scalability but also reduces the flexibility of the model compared to multivariate Hawkes process-based models that specifically encourage reciprocity [2, 7]. Additionally, our estimation procedure uses unregularized spectral clustering to match our theoretical analysis in Section 4. We note that regularized versions of spectral clustering [26, 27, 30, 34, 37] have been found to perform better empirically and would likely improve the community detection accuracy in the CHIP model. Methods that jointly estimate the community structure and Hawkes process parameters, such as the local search and variational inference approaches explored in Junuthula et al. [9] for the Block Hawkes Model could also improve estimation accuracy of both. Also, methods that integrate change point detection with estimation for continuous-time block models could be used to allow for community structure to change over time [8], resulting in more flexible models.

## Broader Impact

Our proposed CHIP model can be applied to analyze any type of timestamped relational event data. In this paper, we considered analysis of mobile phone calls, emails, and user interactions on on-line social networks. However, timestamped relational event data is used in a variety of other disciplines, including financial mathematics, e.g. transactions between traders in financial markets [19]; political science, e.g. military deployments between countries [2, 67]; and sociology, e.g. homicides between gangs in a city [43, 68]. Thus, our CHIP model can have broader impact to society through the advancement of multiple research disciplines.

The CHIP model, like other generative models for dynamic networks, can be used for forecasting, e.g. to predict which nodes are likely to have an event, as well as the number of events during a specified time interval. For some applications, the forecasts may themselves be used to affect decision making. For example, in public policy, crime forecasting can be used for predictive policing, which affects the allocation of police resources to different locations over time. This can have societal benefits, as a recent randomized controlled field trial for predictive policing using Hawkes process models for prediction demonstrated a 7% reduction in crime [69], but also potential for negative consequences like arrests that are biased with respect to minority communities, although such consequences were not observed in the randomized trial [70]. In this paper, we analyzed a publicly available anonymized Facebook on-line social network dataset, so we are not aware of negative consequences that may result from our proposed model.

## Acknowledgments and Disclosure of Funding

This material is based upon work supported by the National Science Foundation grants DMS-1830412, DMS-1830547, and IIS-1755824.

## Footnotes

[1]Code available on GitHub: `https://github.com/IdeasLabUT/CHIP-Network-Model`.

[2]Experiments were run on a workstation with 2 Intel Xeon 2.3 GHz CPUs with a total of 36 cores.

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
