[Supplementary Material]

# Supplemental Material: CHIP: A Hawkes Process Model for Continuous-time Networks with Scalable and Consistent Estimation

Makan Arastuie
EECS Department, University of Toledo
makan.arastuie@rockets.utoledo.edu

Subhadeep Paul
Department of Statistics, The Ohio State University
paul.963@osu.edu

Kevin S. Xu
EECS Department, University of Toledo
kevin.xu@utoledo.edu

# A   Additional Details on Estimation Procedure

## A.1   Community Detection

The spectral clustering algorithm for directed networks that we consider in this paper is shown in Algorithm A.1. It can be applied either to the weighted adjacency (count) matrix $N$ or the unweighted adjacency matrix $A$, where $A_{ij} = 1\{N_{ij} > 0\}$ and $1\{\cdot\}$ denotes the indicator function of the argument. This algorithm is used for the community detection step in our proposed CHIP estimation procedure. For undirected networks, which we use for the theoretical analysis in Section 4, spectral clustering is performed by running k-means clustering on the rows of the *eigenvector* matrix of $N$ or $A$, not the rows of the concatenated singular vector matrix.

## A.2   Estimation of Hawkes process parameters

Ozaki (1979) derived the log-likelihood function for Hawkes processes with exponential kernels, which takes the form:

$$\log \mathcal{L} = -\mu T + \sum_{q=1}^{l} \frac{\alpha}{\beta}\{e^{-\beta(T-t_q)} - 1\} + \sum_{q=1}^{l} \log(\mu + \alpha w(q)) \tag{A.1}$$

where $w(q) = \sum_{q':t_{q'}<t_q} e^{-\beta(t_q - t_{q'})}$. Moreover, $w(q)$ can be computed recursively using $w(q) = e^{-\beta(t_q - t_{q-1})}(1 + w(q - 1))$, with the added base case of $w(1) = 0$, which drops the double summation in the last term and decreases the computational complexity of the log-likelihood from $\mathcal{O}(l^2)$ to $\mathcal{O}(l)$ (Laub et al., 2015). The three parameters $\mu, \alpha, \beta$ can be estimated by maximizing (A.1) using standard numerical methods for non-linear optimization (Nocedal & Wright, 2006).

In our CHIP model, we have separate $(\mu, \alpha, \beta)$ parameters for each block pair $(a, b)$. We provide closed-form equations for estimating $m_{ab} = \alpha_{ab}/\beta_{ab}$ and $\mu_{ab}$ in (2). To separately estimate the $\alpha_{ab}$

---

**Algorithm A.1** Spectral clustering algorithm for community detection in directed networks

   **Input:** Adjacency Matrix $N$, number of blocks $k$

   **Result:** Estimated block assignments $\hat{C}$

  1: Compute singular value decomposition of $N$

  2: $\hat{\Sigma} \leftarrow$ diagonal matrix of $k$ largest singular values of $N$

  3: $\hat{U}, \hat{V} \leftarrow$ left and right singular vectors of $N$ corresponding to $k$ largest singular values

  4: $\hat{Z} \leftarrow$ concatenate($\hat{U}, \hat{V}$)

  5: Normalize the magnitude of each row of $\hat{Z}$ to 1

  6: $\hat{C} \leftarrow$ k-means clustering on rows of $\hat{Z}$

  7: **return** $\hat{C}$

---

and $\beta_{ab}$ parameters, we replace $\alpha_{ab} = \beta_{ab} m_{ab}$ in the exponential Hawkes log-likelihood (A.1) for block pair $(a, b)$ to obtain

$$
\log \mathcal{L}(\beta_{ab} | C, [\mathbf{t}_{ij}]_{i,j=1}^n) = \sum_{i,j:C_{ia}=1, C_{jb}=1} \left\{ -\mu_{ab} T \right.
$$
$$
\left. + \sum_{q=1}^{N_{ij}} m_{ab} \{ e^{-\beta_{ab}(T - t_{ij}^q)} - 1 \} + \sum_{q=1}^{N_{ij}} \log(\mu_{ab} + \beta_{ab} m_{ab} w_{ij}(q)) \right\} \quad \text{(A.2)}
$$

where $w_{ij}(q) = \sum_{q': t_{ij}^{q'} < t_{ij}^q} e^{-\beta_{ab}(t_{ij}^q - t_{ij}^{q'})}$ for $q \geq 2$ and $w_{ij}(1) = 0$. We substitute in the estimates for $m_{ab}$ and $\mu_{ab}$ from (2). Then the log-likelihood (A.2) is purely a function of $\beta_{ab}$ and can be maximized using a standard scalar optimization or line search method. In our experiments, we perform the line search using SciPy's function `minimize_scalar(method="bounded")`.

# B   Additional Theoretical Analysis of Estimators

We present additional results on estimation of community assignments in Section B.1 along with proofs and discussions. We then provide proofs of our results for estimated Hawkes process parameters in Section B.2 along with discussions on obtaining confidence intervals for the estimated Hawkes process parameters.

## B.1   Estimated Community Assignments

We define the notation $Y \sim \text{CHIP}(C, n, k, \mu, \alpha, \beta)$ to denote that relational event matrix $Y$ is generated from a CHIP model with $n$ nodes, $k$ blocks, community assignment matrix $C$ and Hawkes process parameter matrices $(\mu, \alpha, \beta)$. To characterize the misclustering rate of a spectral clustering algorithm applied to $N$, we define the following quantities. Let $\lambda_{\min}(E[N])$ denote the minimum in absolute value non-zero eigenvalue of the matrix $E[N]$. Define

$$
s = \sqrt{T} \max_a \sqrt{ \sum_b |b| \frac{\mu_{ab}}{(1 - \alpha_{ab}/\beta_{ab})^3} }, \quad \text{(B.1)}
$$

$$
s_1 = \sqrt{T} \max_{a,b} \sqrt{ \frac{\mu_{ab}}{(1 - \alpha_{ab}/\beta_{ab})^3} }. \quad \text{(B.2)}
$$

Then we have the following upper bound on the misclustering error rate.

**Theorem B.1.** *Let* $Y \sim CHIP(C, n, k, \mu, \alpha, \beta)$. *Then, with probability at least* $1 - 1/n$, *the misclustering error rate for spectral clustering on the weighted adjacency matrix* $N$ *at time* $T \to \infty$ *is*

$$r \le 64(2 + \epsilon_1)|a|_{\max} k \frac{\left\{(1 + \epsilon)(2s + \frac{6}{\log(1+\epsilon)}s_1 \sqrt{\log n}) + s_1 \sqrt{\log n}\right\}^2}{n(\lambda_{\min}(E[N])^2},$$

*where* $0 < \epsilon < 1/2$ *and* $\epsilon_1 > 0$ *are constants.*

Theorem B.1 provides an upper bound to the error rate of spectral clustering on the weighted adjacency matrix $N$ in the setting $T \to \infty$. Note that the assumption of $T \to \infty$ does not preclude us from being able to analyze scenarios where the network is sparse since the expected number of events between a pair of nodes $\nu_{ab}$ can be made constant or even $o(1)$ by setting $\frac{\mu_{ab}}{1 - \alpha_{ab}/\beta_{ab}} = O(1/T)$ and $\frac{\mu_{ab}}{1 - \alpha_{ab}/\beta_{ab}} = o(1/T)$ respectively.

It is also possible to obtain an expression for the mean as a function of $T$ without the assumption of $T \to \infty$ using stochastic differential equations (Laub et al., 2015; Da Fonseca & Zaatour, 2014). In particular, if we substitute the starting intensity $\lambda_0 = \mu$, i.e., the process starts with baseline intensity as we have assumed throughout, and the starting number of events $N_0 = 0$, then from the result of Da Fonseca & Zaatour (2014) and Daw & Pender (2018),

$$E[N_{ij}] = \frac{\mu_{ab} T}{1 - \alpha_{ab}/\beta_{ab}} - \frac{\mu_{ab} \alpha_{ab} \left[1 - e^{-(\beta_{ab} - \alpha_{ab})T}\right]}{(\beta_{ab} - \alpha_{ab})^2}.$$

We note that there is a small negative correction term to the asymptotic mean, since $\mu_{ab}, \alpha_{ab}, \beta_{ab}, T$ are all non-negative. The effect of this term is negligible as $T \to \infty$, so we ignore it.

We now present an upper bound on the error rate for communities (analogous to Theorem B.1) estimated from the unweighted adjacency matrix $A$. For a pair of nodes $(i, j)$ such that $c_i = a$ and $c_j = b$, we have $E[A_{ij}] = E[1\{N_{ij} > 0\}] = P(N_{ij} > 0) = 1 - e^{(-\mu_{ab}T)}$. Now $A$ is a $n \times n$ symmetric matrix whose elements $A_{ij}$ are independent Bernoulli random variables with mean $E[A_{ij}]$. Let $\Delta = \max\{n \max_{i,j} E[A_{ij}], c_0 \log n\}$ for some constant $c_0$, and note that $n \max_{i,j} E[A_{ij}] = n \max(1 - \exp(-\mu_{ab}T)) = n(1 - \exp(-\mu_{\max}T))$, where $\mu_{\max} = \max_{a,b} \mu_{ab}$. Further, let $\lambda_{\min}(E[A])$ denote the minimum in absolute value non-zero eigenvalue of the matrix $E[A]$ and $|a|_{\max}$ denote the size of the largest community. Then we have the following upper bound on the error rate of spectral clustering performed on $A$.

**Theorem B.2.** *Let* $Y \sim CHIP(C, n, k, \mu, \alpha, \beta)$. *Then, with probability at least* $1 - n^{-r}$, *the misclustering error rate for spectral clustering on the binary adjacency matrix* $A$ *at time* $T$ *is*

$$r \le 64(2 + \epsilon) \frac{|a|_{\max} k c \Delta}{n(\lambda_{\min}(E[A]))^2},$$

*where* $\epsilon > 0$ *is a constant and* $c > 0$ *is a constant dependent on* $c_0$ *and* $r$.

### B.1.1 Simplified Special Case

The upper bounds on the error rates in Theorems B.1 and B.2 are not very informative in terms of their dependencies on key model parameters. In Section 4.1, we considered a simplified special case that allowed us to simplify the constants in Theorem B.1, resulting in Theorem 1, which bounds the misclustering error rate on the weighted adjacency matrix $N$. Similarly, we have the following result for spectral clustering using the unweighted adjacency matrix $A$.

**Theorem B.3.** *Let* $Y \sim CHIP(C, n, k, \mu_1, \alpha_1, \beta_1, \mu_2, \alpha_2, \beta_2)$. *The misclustering error rate for spectral clustering on the binary adjacency matrix $A$ at time $T$ is*

$$r \lesssim \frac{k^2}{n} \frac{1 - \exp(-\mu_1 T)}{(\exp(-\mu_2 T) - \exp(-\mu_1 T))^2}. \tag{B.3}$$

*If further we assume $\mu_1 \asymp \mu_2 \asymp o(1/T)$, such that $\mu_1 T = o(1)$ and $\mu_2 T = o(1)$, then we have*

$$r \lesssim \frac{nT\mu_1}{(n/k)^2(\mu_1 - \mu_2)^2 T^2} \asymp \frac{k^2}{nT} \frac{\mu_1}{(\mu_1 - \mu_2)^2}, \tag{B.4}$$

*whereas, for $\mu_1 \asymp \mu_2 \asymp \omega(1/T)$, such that $\mu_1 T \to \infty$ and $\mu_2 T \to \infty$, then the upper bound for the misclustering rate in Theorem B.2 goes to 1.*

We note that if the parameters are kept constant as a function of $T$, then $\mu_1 T \to \infty$ and $\mu_2 T \to \infty$. Consequently, without $k$ and $n$ changing the upper bound on the error rate for the unweighted adjacency matrix in Theorem B.2 explodes and becomes close to 1, making the upper bound guarantee useless. While this result might be a drawback of the upper bound result itself, we note that unbounded error makes sense because in this regime almost all node pairs have at least one communication with high probability. Hence the unweighted adjacency matrix has a 1 in almost all entries, and the community structure cannot be detected from this matrix. In that case, we predict that using the weighted adjacency matrix $N$ can lead to smaller error. Theorem 1 provides the corresponding upper bound for error rate for $N$.

The density of the aggregate adjacency matrix is governed by the parameters of the CHIP model. Hence, to further characterize the dependence of the $\mu$ parameters on the number of nodes $n$ and time $T$ in the network, assume $\mu_1 = c_1 \frac{1}{f(n)g(T)}$ and $\mu_2 = c_2 \frac{1}{f(n)g(T)}$, where $c_1$ and $c_2$ are constants that do not depend on $n$ or $T$. Also assume $1 - \alpha_1/\beta_1$ and $1 - \alpha_2/\beta_2$ do not depend on $n$ and $T$. Then the upper bound on the error rate becomes $r \lesssim \frac{k^2 f(n)g(T)}{nT(c_1-c_2)^2}$. Now we note that consistent community detection is possible as long as $k = o\left(\frac{\sqrt{nT}|c_1-c_2|}{f(n)g(T)}\right)$. For example, if we set $g(T) \asymp T$ and $f(n) = \frac{n}{\log n}$, such that $\mu_1 \asymp \mu_2 \asymp \frac{\log n}{nT}$, then the expected number of events between a node pair is $O(\frac{\log n}{n})$. In that case, $r(T) \lesssim \frac{k^2}{\log n(c_1-c_2)^2}$, and consistent community detection is possible as long as $k = o(\sqrt{\log n}|c_1 - c_2|)$.

A second example is where we set $g(T) \asymp 1$ and $f(n) = \frac{n}{\log n}$, such that $\mu_1 \asymp \mu_2 \asymp \frac{\log n}{n}$. The expected number of events between a vertex pair is then $O(\frac{T \log n}{n})$ and total expected number of events in the whole network is $O(nT \log n)$. In that case $r \lesssim \frac{k^2}{T \log n(c_1-c_2)^2}$, and consistent community detection is possible as long as $k = o(\sqrt{T \log n}|c_1 - c_2|)$.

### B.1.2 Comparison Between Weighted and Unweighted Adjacency Matrices

We compare the bounds on the error rates in unweighted and weighted adjacency matrices in Theorems B.3 and 1 in the sparse regime where $\mu_1 T$ and $\mu_2 T$ are small such that we can apply the Taylor series approximation. From Theorem B.3, we have the error rate using the unweighted adjacency matrix is upper bounded by $\frac{k^2}{nT} \frac{\mu_1}{(\mu_1-\mu_2)^2}$, while the error rate for the weighted adjacency matrix is upper bounded by

$$\frac{k^2}{nT} \frac{\frac{\mu_1}{(1-m_1)^3} + \frac{\mu_2}{(1-m_2)^3}}{\left(\frac{\mu_1}{(1-m_1)} - \frac{\mu_2}{(1-m_2)}\right)^2}.$$

We can make the following comparison comments on the basis of these upper bounds.

1. If $m_1 = m_2 = m$ such that the community structure is expressed only through $\mu_1$ and $\mu_2$, then the error for the weighted adjacency matrix is bounded by $\frac{k^2}{nT} \frac{\mu_1 + \mu_2}{(\mu_1 - \mu_2)^2} \frac{1}{1-m}$. This upper bound is higher than the corresponding upper bound for spectral clustering in unweighted adjacency matrix indicating a possible advantage of using the unweighted adjacency matrix.

2. If $\mu_1 = \mu_2$ such that the community structure is expressed purely through $\alpha, \beta$, then the error for the unweighted case is unbounded. However, the error for the weighted case can still be bounded, indicating a possible advantage of the weighted adjacency matrix.

### B.1.3 Proofs

We begin with the proofs of Theorems B.1 and B.2 for spectral clustering applied to the weighted and unweighted adjacency matrices, respectively, in the general CHIP model. We then present the proofs of Theorems 1 and B.3 for the simplified special case.

**Proof of Theorem B.1**

*Proof.* We start with the following result.

**Lemma B.1.** *Let $Y \sim CHIP(C, n, k, \mu, \alpha, \beta)$. Let $N$ denote the weighted adjacency matrix obtained by aggregating $Y$ at time $T \to \infty$. Then, with probability at least $1 - 1/n$, we have*

$$\|N - E[N]\|_2 \leq (1 + \epsilon) \left\{ 2s + \frac{6}{\log(1 + \epsilon)} s_1 \sqrt{\log n} \right\} + 2s_1 \sqrt{\log n}, \tag{B.5}$$

*where $0 < \epsilon < 1/2$ is a constant, and the terms $s$ and $s_1$ are as defined in (B.1) and (B.2), respectively.*

We present the proof of this lemma following the proof of this theorem.

Since $E[N]$ can also be written in the form of a stochastic block model as $E[N] = C\nu C^T$, we can use the same arguments as in the proof of the previous result. Using the Davis-Kahan Theorem (Davis & Kahan, 1970; Stewart & Sun, 1990), we have the following bound:

$$r \leq \frac{1}{n} |a|_{\max} 8(2 + \epsilon_1) \|\hat{U} - C(C^T C)^{-1/2} \mathcal{O}\|_F^2$$

$$\leq 64(2 + \epsilon_1) \frac{|a|_{\max} k \|N - E[N]\|_2^2}{n(\lambda_{\min}(E[N])^2}, \tag{B.6}$$

Combining (B.5) and (B.6), we arrive at the desired result.

$\square$

**Proof of Lemma B.1**

*Proof.* We note that $N_{ij}$ is asymptotically normal (Theorem 4 of Hawkes & Oakes (1974)) as $T \to \infty$, i.e.

$$N_{ij} | (C_{ia} = 1, C_{jb} = 1) \sim \mathcal{N}(\nu_{ab}, \sigma_{ab}^2).$$

Then $(N - E[N])$ is a $n \times n$ symmetric matrix with elements $(N - E[N])_{ij} = g_{ij} \sigma_{ij}$, where $g_{ij}; i \geq j$ are i.i.d $\mathcal{N}(0, 1)$ and $\sigma_{ij}$ is the standard deviation of $N_{ij}$ given before.

We will use Corollary 3.9 in Bandeira & van Handel (2016). In the notation of Bandeira & van Handel (2016), we set $\sigma = s$, $\sigma^* = s_1$ and let $t = 2s_1 \sqrt{\log n}$. Then for any $0 < \epsilon < 1/2$, we have

$$P \left( \|N - E[N]\|_2 \geq (1 + \epsilon)\{2s + \frac{6}{\log(1 + \epsilon)} s_1 \sqrt{\log n}\} + 2s_1 \sqrt{\log n} \right) \leq \exp(-\log n).$$

Since $\exp(-\log n) = 1/n$, one can then take the probability of the complement, which completes the proof. $\square$

**Proof of Theorem B.2**

*Proof.* We note that the matrix $A$ is an adjacency matrix with independent entries. Further $n \max_{ij} E[A_{ij}] \le \Delta$ and $\Delta \ge c_0 \log n$ by definition. Then by Theorem 5.2 of Lei & Rinaldo (2015), we have with probability at least $1 - n^{-r}$,

$$\|A - E[A]\|_2 \le c\sqrt{\Delta}, \tag{B.7}$$

where $c$ is a constant dependent on $c_0$ and $r$.

Since $E[A]$ can be written in the form of a stochastic block model as $E[A] = C(1 - \exp(\mu T))C^T$, we can use known results in the SBM literature. Let $\hat{U}_{n \times k}$ denote the $n \times k$ matrix whose columns are the top $k$ eigenvectors of the matrix $A$. By Lemma 3.1 of Rohe et al. (2011), the matrix of eigenvectors corresponding to the largest $k$ non-zero eigenvalues of the matrix $E[A]$ is $C(C^T C)^{-1/2}\mathcal{O}$ for some $k \times k$ orthogonal matrix $\mathcal{O}$. Then we have the following relationship for the difference between matrices of population eigenvectors (those of $E[A]$) and sample eigenvectors (those of $A$) and the misclustering error rate of community detection by applying $(1 + \epsilon)$ approximate $k$-means algorithm to those matrices (Pensky & Zhang, 2019):

$$r \le \frac{1}{n}|a|_{\max}8(2 + \epsilon)\|\hat{U} - C(C^T C)^{-1/2}\mathcal{O}\|_F^2. \tag{B.8}$$

Next we use the Davis-Kahan Theorem (Davis & Kahan, 1970; Stewart & Sun, 1990) that relates perturbation of matrices to perturbation of eigenspaces of those matrices. Then we have the following bound on the misclustering rate (also see Lemma 5.1 of Lei & Rinaldo (2015)):

$$r \le 64(2 + \epsilon)\frac{|a|_{\max}k\|A - E[A]\|_2^2}{n(\lambda_{\min}(E[A])^2}. \tag{B.9}$$

Combining (B.7) and (B.9), we arrive at the desired result.

$\square$

Next, we present the proofs of the theorems for the simplified special case with $k$ equivalent communities.

**Proof of Theorem 1**

*Proof.* Under the simplified model we have

$$E[N] = C\left((\nu_1 - \nu_2)T I_k + \nu_2 T \mathbf{1}_k \mathbf{1}_k^T\right)C^T.$$

As before all communities have the same number of nodes, i.e., $|a| = \frac{n}{k}$ for all $a$, and $|a|_{\max} = \frac{n}{k}$. Then by Rohe et al. (2011), $\mathbf{1}_k$ is an eigenvector corresponding to the eigenvalue $\frac{n}{k}(\nu_1 - \nu_2)T + n\nu_2 T$, and the remaining non-zero eigenvalues are of the form $\frac{n}{k}(\nu_1 - \nu_2)T$. Since $n\nu_2 > 0$, the smallest non-zero eigenvalue

$$\lambda_{\min}(E[N]) = \frac{n}{k}(\nu_1 - \nu_2)T.$$

The upper bound from Theorem B.1 can also be simplified further under this model. We have

$$s = \sqrt{T}\sqrt{\frac{n}{k}\sigma_1^2 + \frac{(k-1)n}{k}\sigma_2^2} \asymp \sqrt{\frac{nT}{k}}\sqrt{\sigma_1^2 + (k-1)\sigma_2^2} \asymp \sqrt{nT}\sigma_1,$$

and

$$s_1 = \sqrt{T}\sigma_1,$$

and consequently,

$$(1 + \epsilon)\left(2s + \frac{6}{\log(1 + \epsilon)}s_1\sqrt{\log n}\right) + 2s_1\sqrt{\log n} \asymp \sqrt{T}\sigma_1\left(\sqrt{n} + \sqrt{\log n}\right)$$
$$\lesssim \sqrt{T}\sigma_1\sqrt{n}.$$

Substituting these quantities into Theorem B.1 completes the proof.

$\square$

**Proof of Theorem B.3**

*Proof.* Under the simplified model all communities have the same number of nodes, i.e., $|a| = \frac{n}{k}$ for all $a$, and consequently $|a|_{\max} = \frac{n}{k}$. Further, we can write

$$E[A] = C\left((\exp(-\mu_2 T) - \exp(-\mu_1 T))I_k + (1 - \exp(-\mu_2 T)\mathbf{1}_k\mathbf{1}_k^T\right)C^T,$$

where $I_k$ is the $k$-dimensional identity matrix, and $\mathbf{1}_k$ is the $k$-dimensional vector of all 1's. Then by Rohe et al. (2011), $\mathbf{1}_k$ is an eigenvector corresponding to the eigenvalue $\frac{n}{k}(\exp(-\mu_2 T) - \exp(-\mu_1 T)) + n(1 - \exp(-\mu_2 T))$, and the remaining non-zero eigenvalues are of the form $\frac{n}{k}(\exp(-\mu_2 T) - \exp(-\mu_1 T))$. Since $n(1 - \exp(-\mu_2 T)) > 0$, the smallest in absolute value non-zero eigenvalue of $E[A]$ is then,

$$\lambda_{\min}(E[A]) = \frac{n}{k}(\exp(-\mu_2 T) - \exp(-\mu_1 T)).$$

Also, under this setting, the numerator in the upper bound from Theorem B.2 becomes

$$\Delta = n(1 - \exp(-\mu_1 T)).$$

Substituting these quantities into Theorem B.2, we arrive at (B.3), the first statement of the theorem.

If we further assume that $\mu T$ is small then we can make some further simplifications using the Taylor series expansion of $\exp(-x)$ near $x = 0$. In this case,

$$\lambda_{\min} \asymp \frac{n}{k}(\mu_1 - \mu_2)T,$$

and

$$\Delta \asymp n\mu_1 T.$$

Substituting these quantities into Theorem B.2, we arrive at (B.4), the second statement of the theorem, which completes the proof. $\square$

## B.2 Estimated Hawkes Process Parameters

### B.2.1 Confidence Intervals

We derive confidence intervals for $m$ using Theorem 2 and the following result readily obtained using the Law of Large numbers: $\bar{N}_{ab} \xrightarrow{p} \mu_{ab}$. A $(1 - \theta) * 100\%$ Bonferroni-corrected (due to multiple comparisons) simultaneous confidence interval for all $k^2$ parameters $m_{ab}$ is

$$\hat{m}_{ab} \pm z_{(1 - \frac{\theta}{2k^2})}\sqrt{\frac{1}{4n_{ab}\bar{N}_{ab}}}. \tag{B.10}$$

The confidence intervals on $m_{ab}$ are particularly appealing to detect the "burstiness" of the network dynamics by testing the hypothesis $m_{ab} > 0$ for a block pair $(a, b)$.

For the $\mu$ parameters, we are more interested in confidence intervals for pairwise differences between block pairs to identify whether the block pairs differ in their baseline event rates. Therefore, we build the following pairwise confidence intervals for all $2k(k-1)$ pairwise differences:

$$(\hat{\mu}_{ab} - \hat{\mu}_{ac}) \pm z_{(1 - \frac{\theta}{4(k-1)k})} \frac{1}{T} \sqrt{\frac{9}{4} \left( \frac{\bar{N}_{ab}}{n_{ab}} + \frac{\bar{N}_{ac}}{n_{ac}} \right)}. \tag{B.11}$$

Note even though the random variables $\hat{m}_{ab}$ and $\hat{\mu}_{ab}$ are dependent across block pairs due to the spectral clustering step, the Bonferroni correction is still going to give a conservative (wide) interval with a simultaneous confidence coverage at least $1 - \theta$.

### B.2.2 Proofs

**Proof of Theorem 2**

*Proof.* First, using the Central Limit Theorem and Law of Large Numbers, we have

$$\bar{N}_{ab} \overset{d}{\to} \mathcal{N} \left( \nu_{ab}, \frac{\sigma_{ab}^2}{n_{ab}} \right) \text{ and } S_{ab}^2 \overset{p}{\to} \sigma_{ab}^2, \qquad \text{as } n_{ab} \to \infty.$$

Then by Slutsky's theorem (Lehmann, 2004) we have,

$$\frac{\bar{N}_{ab}}{S_{ab}^2} \overset{d}{\to} \mathcal{N} \left( \frac{\nu_{ab}}{\sigma_{ab}^2}, \frac{1}{\sigma_{ab}^2 n_{ab}} \right) \Leftrightarrow \sqrt{n_{ab}} \left( \frac{\bar{N}_{ab}}{S_{ab}^2} - \frac{\nu_{ab}}{\sigma_{ab}^2} \right) \overset{d}{\to} \mathcal{N} \left( 0, \frac{1}{\sigma_{ab}^2} \right).$$

Finally, we will apply the delta method (See Theorem 2.5.2 of Lehmann (2004)) on the random variable $X = \frac{\bar{N}_{ab}}{S_{ab}^2}$ with the function $g(x) = 1 - \sqrt{x}$. Note that $g'(x) = \frac{1}{2\sqrt{x}}$. Then we can compute $g'\left( \frac{\nu_{ab}}{\sigma_{ab}^2} \right) = \frac{\sigma_{ab}}{2\sqrt{\nu_{ab}}}$. Then we have

$$\sqrt{n_{ab}} \left( \hat{m}_{ab} - \left( 1 - \sqrt{\frac{\nu_{ab}}{\sigma_{ab}^2}} \right) \right) \overset{d}{\to} \mathcal{N} \left( 0, \frac{1}{4\nu_{ab}} \right).$$

Next we derive the asymptotic distribution for $\hat{\mu}_{ab}$. We first apply the delta method to the random variable $\bar{N}_{ab}$ with the function $g(x) = x^{3/2}$. Clearly, $g'(x) = \frac{3}{2}\sqrt{x}$, such that $g'(\nu_{ab}) = \frac{3}{2}\sqrt{\nu_{ab}}$. Then we have

$$\sqrt{n_{ab}}((\bar{N}_{ab})^{3/2} - (\nu_{ab})^{3/2}) \overset{d}{\to} \mathcal{N} \left( 0, \frac{9}{4}\nu_{ab}\sigma_{ab}^2 \right).$$

Applying Slutsky's theorem, we then have

$$\sqrt{n_{ab}} \left( \frac{(\bar{N}_{ab})^{3/2}}{S_{ab}} - \frac{(\nu_{ab})^{3/2}}{\sigma_{ab}} \right) \overset{d}{\to} \mathcal{N} \left( 0, \frac{9}{4}\nu_{ab} \right).$$

$\square$

**Proof of Theorem 3**

Let $\bar{C}$ and $\hat{C}$ denote the true and estimated community assignment matrices respectively. Define $\bar{H} = \bar{C}(\bar{C}^T\bar{C})^{-1/2}$ and $\hat{H} = \hat{C}(\hat{C}^T\hat{C})^{-1/2}$, such that $\bar{H}^T\bar{H} = \hat{H}^T\hat{H} = I$.

We have

$$E[N] = \bar{C}\nu\bar{C}^T$$

Then

$$(\bar{C}^T\bar{C})^{1/2}\nu(\bar{C}^T\bar{C})^{1/2} = (\bar{C}^T\bar{C})^{-1/2}\bar{C}^T E[N]\bar{C}(\bar{C}^T\bar{C})^{-1/2} = \bar{H}^T E[N]\bar{H}.$$

Instead, the estimate for $\nu$ we get using estimated community assignment matrix $\hat{C}$ applied to $N$ is

$$(\hat{C}^T\hat{C})^{1/2}\hat{\nu}(\hat{C}^T\hat{C})^{1/2} = \hat{H}^T N\hat{H}$$

Note that $(\hat{C}^T\hat{C})$ and $(\bar{C}^T\bar{C})$ are $k \times k$ diagonal matrices whose $q$th diagonal element represents the number of vertices that are part of the $q$th community. Next we make a key assumption—the sizes of the communities from the estimated community partition are similar to the true community sizes. In particular, we assume that the size of each of the $k$ communities in the true and estimated partition is $O(\frac{n}{k})$. Therefore, the difference

$$(\hat{C}^T\hat{C})^{1/2}\hat{\nu}(\hat{C}^T\hat{C})^{1/2} - (\bar{C}^T\bar{C})^{1/2}\nu(\bar{C}^T\bar{C})^{1/2} \asymp \frac{n}{k}(\hat{\nu} - \bar{\nu}).$$

Now we have

$$\begin{aligned}
\frac{n}{k}(\hat{\nu} - \nu) &= \hat{H}^T N\hat{H} - \bar{H}^T E[N]\bar{H} \\
&= \hat{H}^T N\hat{H} - \hat{H}^T E[N]\hat{H} + \hat{H}^T E[N]\hat{H} - \bar{H}^T E[N]\bar{H} \\
&= \hat{H}^T (N - E[N])\hat{H} + \{\hat{H}^T E[N](\hat{H} - \bar{H}) + (\hat{H} - \bar{H})^T E[N]\bar{H}\}
\end{aligned}$$

We also note that

$$\|\bar{H}\|_2 \leq \sqrt{\lambda_{\max}(\bar{H}^T\bar{H})} = 1,$$

Note by assumption, $\sqrt{n_{ab}} \asymp \frac{n}{k}$. Now,

$$\begin{aligned}
\sum_{ab} n_{ab}(\hat{\nu} - \nu)_{ab}^2 &\asymp \left(\frac{n}{k}\right)^2 \|\hat{\nu} - \nu\|_F^2 \\
&\leq (\|\hat{H}^T(N - E[N])\hat{H}\|_F + 2\|\hat{H}^T E[N](\hat{H} - \bar{H})\|_F)^2 \\
&\leq 2k\left(\|N - E[N]\|_2^2 + 4\|E[N]\|_2^2 \frac{\|N - E[N]\|_2^2}{\lambda_{min}^2(N)}\right)
\end{aligned}$$

In the notation of Theorem 1,

$$\lambda_{\min}(N) = \frac{n}{k}(\nu_1 - \nu_2)T.$$

Also, using the upper bound in terms of expectation (instead of the in probability upper bound) from Theorem 1 we have

$$E[\|N - E[N]\|_2] \lesssim \sqrt{nT}\sigma_1, \quad \|E[N]\|_2 \lesssim n\nu_1 T.$$

Therefore,

$$E\left[\sum_{ab} n_{ab}(\hat{\nu} - \nu)_{ab}^2\right] \lesssim knT\sigma_1^2 + k\frac{n^2\nu_1^2 T^2 nT\sigma_1^2 k^2}{n^2(\nu_1 - \nu_2)^2 T^2} \lesssim knT\sigma_1^2 + \frac{k^3 nT\sigma_1^2\nu_1^2}{(\nu_1 - \nu_2)^2}.$$

And consequently the sum of the weighted mean squared errors is,

$$\sum_{ab} n_{ab} E[(\hat{\nu} - \nu)^2_{ab}] \lesssim knT \max\left\{\sigma_1^2, \frac{k^2 \sigma_1^2 \nu_1^2}{(\nu_1 - \nu_2)^2}\right\}$$

Noting that $\sum_{ab} n_{ab} \asymp n^2$, the average MSE of estimating $\nu_{ab}$ is then asymptotically

$$\frac{kT}{n} \max\left\{\sigma_1^2, \frac{k^2 \sigma_1^2 \nu_1^2}{(\nu_1 - \nu_2)^2}\right\} \text{ or } \frac{T}{\sqrt{n_{ab}}} \max\left\{\sigma_1^2, \frac{k^2 \sigma_1^2 \nu_1^2}{(\nu_1 - \nu_2)^2}\right\}$$

For comparison, the weighted sum of MSEs in estimating $\nu_{ab}$, using the estimator $\bar{N}_{ab}$ when the community structure is known (from Theorem 2) is

$$\sum_{ab} n_{ab} E[(\bar{N}_{ab} - \nu_{ab})^2] = \sum_{ab} \sigma_{ab}^2 = k^2 T \sigma_1^2,$$

and average MSE is asymptotically

$$\frac{k^2 T \sigma_1^2}{n^2} \text{ or } \frac{T \sigma_1^2}{n_{ab}}.$$

# C   Additional Experiments

We present two additional simulation experiments to analyze the effects of various parameters of the CHIP model on the accuracy of spectral clustering and to compare spectral clustering using weighted and unweighted adjacency matrices in detecting the ground truth community structure in simulated networks. We then present additional details and analyses for our real network dataset experiments.

## C.1   Simulation Experiments

### C.1.1   Community Detection with Varying $n$

We simulate networks from the simplified CHIP model described in Section B.1.1 with $k = 4$ communities, duration $T = 400$, and a growing number of nodes $n$. We estimate community assignments of nodes using both the weighted adjacency (count) matrix $N$ and unweighted adjacency matrix $A$.

First, we choose parameters $\mu_1 = 0.002$, $\mu_2 = 0.001$, $\alpha_1 = \alpha_2 = 7$, and $\beta_1 = \beta_2 = 8$ so that only $\mu$ is informative. The upper bound on the misclustering error rate using $N$ is worse by a factor of $(1 - m)^{-1} = 8$ compared to using $A$ as discussed in Section B.1.2. The adjusted Rand scores for spectral clustering on both $A$ and $N$ over 100 simulated networks for varying $n$ are shown in Figure C.1(a). The accuracy on $A$ approaches 1 for growing $n$, as expected. The accuracy on $N$ is significantly worse, as predicted by the comparison of the respective upper bounds on the misclustering error rates, and no better than a random community assignment until $n = 512$ nodes.

Next, we choose parameters $\mu_1 = \mu_2 = 0.001$, $\alpha_1 = 0.006$, $\alpha_2 = 0.001$, and $\beta_1 = \beta_2 = 0.008$. so that only $\alpha$ is informative. The error for $A$ is unbounded, while the error for $N$ still follows the upper bound in Corollary 1. As shown in Figure C.1(b), the accuracy on $N$ approaches 1 as $n$ increases, while the accuracy on $A$ is no better than random even for growing $n$, as expected.

(a) Only $\mu$ is informative ($\mu_1 \neq \mu_2$ and $\alpha_1 = \alpha_2$)

(b) Only $\alpha$ is informative ($\mu_1 = \mu_2$ and $\alpha_1 \neq \alpha_2$)

Figure C.1: Mean adjusted Rand scores of spectral clustering on weighted and unweighted adjacency matrices over 100 simulated networks ($\pm$ 2 standard errors). $\beta_1 = \beta_2$ in both cases. Both sets of results agree with upper bounds from Section B.1.2.

### C.1.2 Effects of Diagonal and Off-diagonal $\mu$'s on Community Detection

In Section C.1.1 we observed that community detection will be easier if $\mu$ is informative ($\mu_1 \neq \mu_2$). In this experiment, we will explore two different ways of encoding community information into simulated networks by

1. Scaling up both $\mu_1$ and $\mu_2$, while keeping a fixed $\mu_1 : \mu_2$ ratio.

2. Only scaling up $\mu_1$, allowing for $\mu_1 : \mu_2$ ratio to increase.

Both settings share the same base parameters of $\mu_1 = 0.075$ and $\mu_2 = 0.065$, with $k = 4$ communities and $n = 128$ nodes, a duration of $T = 50$, where $\alpha_1 = \alpha_2 = 0.05$ and $\beta_1 = \beta_2 = 0.08$. These parameters are chosen to create a base network that is nearly impossible for spectral clustering to accurately detect communities. The objective is similar to that of Section C.1.1, where in both settings we perform community detection using spectral clustering on the weighted adjacency of simulated networks, while increasing $\mu_1$ and $\mu_2$ or their ratio. Lastly, we average over the adjusted Rand score of 100 simulations.

As shown in Figure C.2, community detection accuracy increases in both settings as the scalars increase; however, we find that the increase in accuracy occurs for different reasons. In the first setting, Figure C.2(a), where both $\mu$'s are scaled up with a fixed ratio, community detection becomes easier simply because the networks are becoming denser, as shown in the numbers above the bars in Figure C.3, and more information is available. Furthermore, although we keep the $\mu_1 : \mu_2$ ratio fixed, as the scalars increase the difference between the two starts to magnify. On the other hand, as networks become denser and most node pairs start to have at least one interaction, it is only the number of interactions among node pairs that becomes informative. Therefore, spectral clustering on the weighted adjacency matrix continues to result in a high adjusted Rand score, while the adjusted Rand score of spectral clustering on the unweighted adjacency matrix decreases with increasing density, as it is illustrated in Figure C.3. This observation confirms the theoretical prediction made in Theorem B.3. The opposite also holds to some degree. For really sparse networks spectral clustering is more accurate on the unweighted adjacency matrix; however, in Figure C.3 we observe that it loses its advantage as the proportion of node pairs with at least one interaction approaches 0.5 and starts impairing community detection as it passed 0.8.

(a) Scaling both $\mu$'s up, with a fixed $\mu_1 : \mu_2$ ratio

(b) Scaling $\mu_1$ up only, keeping $\mu_2$ fixed

Figure C.2: Adjusted Rand score of spectral clustering on weighted adjacency matrix, averaged over 100 simulated networks ($\pm$ 2 standard errors), while multiplying $\mu_1$ and $\mu_2$ or their ratio by scalars. C.2(a) Scaling up both $\mu_1$ and $\mu_2$, keeping their ratio fixed. C.2(b) Only scaling up $\mu_1$, while keeping $\mu_2$ fixed.

In the second setting, Figure C.2(b), by only scaling up $\mu_1$, the difference between the baseline rate of occurrence of an event between the diagonal and the off-diagonal blocks increases. This increases the signal-to-noise ratio and is a more effective way of encoding community information into a network. This can be observed by comparing the scalars of Figures C.2(a) and C.2(b). Starting from the same base network, a perfect adjusted Rand score is achieved when only $\mu_1$ is scaled up by a factor of 1.4, compared to scaling both $\mu$'s up by a factor of 30.

## C.2 Real Data

### C.2.1 Dataset Descriptions

We consider three real network datasets consisting of timestamped relational events. For each dataset, we normalize the event times to the range $[0, 1{,}000]$.

- MIT Reality Mining (Eagle et al., 2009): Consists of 2,161 phone calls where the start time of each call was used as the event timestamp. This dataset has a "core-periphery" structure, where there is a core group for whom we have all of their communication data and a much larger group of people in the periphery who had contact with the core. We consider calls between pairs of the core 70 callers and recipients. We use the last 661 phone calls as the test set[1].

- Enron (Klimt & Yang, 2004): Consists of 4,000 emails exchanged among 142 individuals. We use the last 1,000 emails as the test set.

- Facebook Wall Posts (Viswanath et al., 2009): Consists of a total of 876,993 wall posts from 46,952 users from September 2004 to January 2009. We consider only posts from a user to

Figure C.3: Adjusted Rand score of spectral clustering on weighted vs. unweighted adjacency matrices, averaged over 100 simulations ($\pm$ 2 standard errors), while multiplying both $\mu_1$ and $\mu_2$ by scalars. The numbers above each bar indicate the average density of simulated networks as the proportion of non-zero entries to the total number of elements in the adjacency matrix. Base model parameters are: $\mu_1 = 7.5 \times 10^{-4}$, $\mu_2 = 3.5 \times 10^{-4}$, $k = 4$, $T = 50$, $n = 256$, $\alpha_1 = \alpha_2 = 0.05$, and $\beta_1 = \beta_2 = 0.08$.

another user's wall so that there are no self-edges. We analyze the largest connected component of the network excluding self loops: 43,953 nodes and 852,833 events. We divide the dataset into train and test sets using a 80%/20% split on the number of events.

### C.2.2 Comparison with Other Models

We find that our proposed CHIP model achieves higher test log-likelihood than the relational event model (REM) (DuBois et al., 2013), block Hawkes model (BHM) (Junuthula et al., 2019), and the spectral clustering with homogeneous Poisson process baseline on the Reality Mining and Enron datasets as shown in Table 1. CHIP and the Poisson baseline were able to scale to the Facebook network, which was two orders of magnitude larger. The local search procedure in the BHM does not scale to such a large network, so we provide fits using only the faster but less accurate spectral clustering procedure. We did not implement the REM so we compare against the reported results in DuBois et al. (2013), which did not include the Facebook data. We note that since all the three models assume the same Poisson process for arrival of events with different rates (which are governed by different set of parameters), the joint distribution of event times has the same form for all three models. Hence the likelihood function of the models are directly comparable. Therefore, the test log-likelihood is a reasonable metric for comparing the fits of the models to the data.

To compute the test log-likelihood for CHIP and BHM, we use the following process. First, we use the estimation procedure explained in Section 3.2 to estimate all CHIP's Hawkes process parameters using the training set (the entire dataset excluding the test set). Next, we calculate the model log-likelihood on the entire dataset and subtract the training log-likelihood from it. The result is then divided by the total number of events in the test set to evaluate the mean log-likelihood per test event, which is the metric used in DuBois et al. (2013). Lastly, if a node in the test set did not appear in the training data, it was automatically assigned to the largest block.

We implemented the BHM by using spectral clustering followed by local search (Junuthula et al., 2019), which they found to achieve the highest adjusted Rand score in simulations compared to just spectral clustering and variational EM. We allowed the local search to converge to a local maximum

Figure C.4: 15 largest singular values of spectral clustering on the weighted adjacency matrix of the Enron dataset. The gap between the $2^{\text{nd}}$ and the $3^{\text{rd}}$ largest singular values led us to select $k = 2$ blocks.

Table C.1: Number of node pairs and events in each block pair of the CHIP model with $k = 2$ in the Enron dataset.

| Block Pair (a, b) | (1, 1) | (1, 2) | (2, 1) | (2, 2) |
|---|---|---|---|---|
| Node Pair Count | 5700 | 5016 | 5016 | 4290 |
| Event Count | 965 | 572 | 1038 | 1425 |

for all values of $k$.

### C.2.3 Exploratory Analysis of Enron Network

Next, we perform model-based exploratory analysis of the Enron network using CHIP. We find a large gap between the $2^{\text{nd}}$ and the $3^{\text{rd}}$ largest singular values of the weighted adjacency matrix as shown in Figure C.4 so we choose a fit with $k = 2$ blocks. The number of node pairs and events in each block pair are shown in Table C.1.

Figure C.5(a) shows the estimated baseline intensity of each block pair. This can be thought of as the rate at which email conversations get started. We observe that $\hat{\mu}_{11}$ is much larger than $\hat{\mu}_{22}$; however block pair $(1, 1)$ only accounts for 965 emails as opposed to 1,425 for block pair $(2, 2)$. Thus, the community structure is not evident only from the differences in the baseline rates $\mu$.

It is only after we consider how bursty interactions are in each block pair, as shown in Figure C.5(b), that we can explain the dynamics of this network. In particular, $\hat{m}_{22}$ is much higher than $\hat{m}_{11}$. In other words, once an email conversation is started in block-pair $(2, 2)$ we can expect more emails to follow, as opposed to more frequent conversations starting in $(1, 1)$, but with less follow-ups. Hence, the combination of $\mu$ and $m$ allows us to observe the community structure, with more edges within block pairs than between, as shown by the values of $\hat{\mu}/(1 - \hat{m})$ in Figure C.5(c).

Table C.2 shows the numerical values for $\hat{m}$ along with their 95% confidence intervals obtained using (B.10), indicating that all block pairs exhibit highly bursty behavior. As previously mentioned, the baseline rates $\hat{\mu}$ are not by themselves indicative of the community structure due to the burstiness of events in all of the blocks. Indeed, when we examine the 95% confidence intervals for pairwise differences between the $\mu$ values for different block pairs using (B.11) shown in Table C.3, all of the confidence intervals include 0.

(a) $\hat{\mu}$  (b) $\hat{m}$  (c) $\hat{\mu}/(1-\hat{m})$

Figure C.5: Estimated CHIP parameters on Enron data, where axis labels of each heatmap denote block index. Each tile corresponds to a block pair where $(a, b)$ denotes row $a$ and column $b$.

Table C.2: Estimated $\hat{m}_{ab} \pm 95\%$ confidence interval from CHIP on the Enron dataset with $k = 2$. All values of $\hat{m}_{ab}$ are statistically significant at the 5% level for the test $m_{ab} > 0$. The high values of $\hat{m}_{ab}$ indicate that interactions in all block pairs are quite bursty.

| Block Pair | 1 | 2 |
|---|---|---|
| 1 | $0.7536 \pm 0.0440$ | $0.7855 \pm 0.0572$ |
| 2 | $0.8126 \pm 0.0424$ | $0.9237 \pm 0.0362$ |

Table C.3: Pairwise difference for unique pairs of diagonal vs. off-diagonal $\hat{\mu}_{a,a} - \hat{\mu}_{a,b} \pm 95\%$ confidence interval of the CHIP model fitted to the Enron dataset with $k = 2$. None of the differences are statistically significant at the 5% level for the test $\hat{\mu}_{a,a} - \hat{\mu}_{a,b} \neq 0$, suggesting that the community structure is not evident from differences in the baseline rates $\mu$.

| Pairwise Differences in $\hat{\mu}$ | |
|---|---|
| $\hat{\mu}_{1,1} - \hat{\mu}_{1,2}$ | $1.724 \times 10^{-5} \pm 2.970 \times 10^{-5}$ |
| $\hat{\mu}_{1,1} - \hat{\mu}_{2,1}$ | $2.929 \times 10^{-6} \pm 3.455 \times 10^{-5}$ |
| $\hat{\mu}_{2,2} - \hat{\mu}_{1,2}$ | $8.650 \times 10^{-7} \pm 4.105 \times 10^{-5}$ |
| $\hat{\mu}_{2,2} - \hat{\mu}_{2,1}$ | $-1.345 \times 10^{-5} \pm 4.468 \times 10^{-5}$ |

(a) Singular values

(b) Block sizes

Figure C.6: Results of spectral clustering on the weighted adjacency matrix of the largest connected component of the Facebook Wall Posts dataset. C.6(a) 100 largest singular values. There is a large gap between the $10^{\text{th}}$ and the $11^{\text{th}}$ largest singular values that leads us to select $k = 10$ blocks. C.6(b) Size of each formed block. Numbers on top of each bar indicate the actual number of nodes in that block.

Figure C.7: Adjacency matrix for Facebook wall post network with rows and columns rearranged to show block structure.

### C.2.4 Exploratory Analysis of Facebook Wall Post Network

Fitting the CHIP model to the largest connected component of the network (excluding self loops) consisting of 43,953 nodes and 852,833 edges required only 141.4 s. Considering the gap between the $10^{\text{th}}$ and the $11^{\text{th}}$ largest singular values of the weighted adjacency matrix of the network as shown in Figure C.6(a), we choose a model with $k = 10$ blocks, resulting in the block sizes depicted in Figure C.6(b).

Figure C.8 shows heatmaps of the fitted CHIP parameters. Although diagonal block pairs have a higher base intensity on average, indicating an underlying assortative community structure, there

(a) $\mu$: base intensity

(b) $m$: $\alpha$ to $\beta$ ratio

(c) $\alpha$: intensity jump size

(d) $\beta$: intensity decay rate

(e) Total number of events

(f) Mean number of events per node pair

Figure C.8: Inferred CHIP parameters on the largest connected component of the Facebook Wall Posts dataset with $k = 10$. Axis labels denote block numbers. Each tile corresponds to a block pair where $(a, b)$ denotes row $a$ and column $b$. Boxed block pairs in the heatmap are discussed in the body text.

are some off-diagonal block pairs with a high $\mu$ such as $(5, 8)$ and $(8, 5)$, as shown in the red boxes in Figure C.8. This illustrates that the CHIP model does not discourage inter-block events. These patterns often occur in social networks, for instance, if there are communities with opposite views on a particular subject.

While the structure of $\mu$ reveals insights on the baseline rates of events between block pairs, the structure of $\alpha$ (Figure C.8(c)) and $\beta$ (Figure C.8(d)) reveal insights on the burstiness of events between block pairs. Note that the structure of the $\alpha$ to $\beta$ ratio $m$ (Figure 4(c)) affects the asymptotic mean number of events in (3). For some block pairs, such as $(3, 10)$, there are very low values of $\alpha$ and $\beta$ indicating the events are closely approximated by a homogeneous Poisson process. There are some block pairs, such as $(2, 8)$ that have a low baseline rate of events but are extremely bursty, which relatively increases the mean number of events per node pair. Both of these block pairs are shown in the blue dashed boxes in Figure C.8. The different levels of burstiness of block pairs cannot be seen from aggregate statistics such as the total number of events (Figure C.8(e)) or even the mean number of events per node pair (Figure C.8(f)).

Unlike the findings of Junuthula et al. (2019), who studied only a subset of the network containing $3,582$ nodes using $k = 2$ blocks, we find that $\alpha$ is not necessarily higher for diagonal blocks as shown in Figure C.8(c). Additionally, even though we do not explicitly model reciprocity between node pairs in our CHIP model, we can nevertheless empirically observe certain reciprocities through the patterns of the estimated $\alpha$ and $\beta$ parameters. We note that the high reciprocity present in social networks is captured by CHIP through the symmetry in all Hawkes process parameters about block pairs. This can be observed in block pairs $(8, 5)$ and $(5, 8)$. In the context of this dataset, a symmetric $\alpha$ and $\beta$ corresponds to the notion that wall posts posted by the people in block 5 on the wall of the people in block 8 will urge people in block 8 to respond, which in turn promotes more wall posts by people in block 5.

Lastly, it is worth noting that fitting the CHIP model to this data set using the unweighted adjacency matrix resulted in a per event log-likelihood of $-10.04$ compared to $-9.61$ for the weighted adjacency matrix on the test data set when using a $80\%/20\%$ train and test split on the events. Thus, this was another reason to use the weighted adjacency matrix besides its aforementioned advantages in previous sections. We note that running spectral clustering on the unweighted adjacency matrix, compared to the weighted adjacency matrix, seemed to detect communities with larger number of intra-block events, while inter-block events were a lot less common.

## Footnotes

[1]We found some inconsistencies between the actual dataset used and its description in DuBois et al. (2013). For a fair comparison, we loaded and preprocessed this dataset using their code available on GitHub: `https://github.com/doobwa/blockrem/blob/master/process/reality.r`.