[Reviews · NeurIPS 2020]

Review 1

Summary and Contributions: The paper introduces Community Hawkes Independent Pairs (CHIP) model for temporal networks with time-stamped links. In the CHIP model, the edge formation between two nodes is governed by a univariate Hawkes process with each event of the Hawkes process indicating an edge. The Hawkes processes for each pair of nodes are independent of each other. The paper focuses on a special case of the Hawkes process with the exponential kernel. Also, the paper considers a stochastic block model formulation, where, the parameters of the Hawkes process are the same if the nodes of the edges belong to the same community. The main contributions of the paper are - (1) Proposing the CHIP model for networks with time-stamped edges. (2) Proposing a method for estimation of the parameters of the CHIP model. (3) Providing theoretical results on error bounds of the proposed estimators. (4) An empirical study of the performance of the estimators as well as application to a large real data set for community detection.

Strengths: (1) Soundness of claim: The paper provides rigorous theoretical justifications of the analytical results presented in the paper. The paper has three main theoretical results - two on the error bounds of the estimators and another on large sample distribution of the estimated parameters. The proofs of the results are well-presented and thorough. The paper also presents a simulation study on the error bounds of the estimators and applications based on community detection to three real datasets and one large Facebook dataset. (2) Significance and novelty: The paper has moderate significance and novelty. (3) Relevance: The paper is relevant and timely. Temporal networks with time-stamped edges are quite common nowadays. The paper provides a simple model for such networks and gives initial methods of analysis. The paper can motivate further works that will be able to better model and build more powerful analysis tools for such temporal networks.

Weaknesses: (1) Soundness of claim: The paper provides analytical results for some special cases of the proposed model. (a) The paper only deals with the estimation of the parameter m = alpha/beta of the Hawkes process theoretically. It does not go into the details of the profile likelihood estimation scheme proposed for estimation of alpha and beta and their theoretical properties. (b) The paper concentrates on dense graphs. The dependence of parameter 'mu' of Hawkes process on the node size 'n' is not discussed in detail. The parameter is considered constant in 'n' and 'T' in the discussions. But, it is clear from the proof structure (which uses the Lei-Rinaldo proof technique directly), that 'mu' can be of the order of log(n)/nT. (c) The estimation of number of communities (k) is not handled in the paper, but it would be good to have at least heuristic methods for this special case. Also in the real large networks, (2) Significance and novelty: The theoretical analysis in the paper is based on established proof techniques. However, the proof techniques have not been used to their full power. Also, with some modifications of the estimators, they can be shown to be consistent for even sparser settings. Also, the main challenge in estimation of parameters of Hawkes process has been bypassed by considering a special parameterization of the Hawkes process. (3) Relevance: The paper is relevant and timely.

Correctness: The analytical proofs of the theoretical results on error bounds of the estimates seem correct based on a cursory reading of the statements and proofs given in the paper and the Supplement. The empirical study is also well-thought. However, it would be good to see the relationship between MSE and the parameters (which are dependent on n and T) of the Hawkes process.

Clarity: The paper is well-written.

Relation to Prior Work: The paper is related to several prior works on community detection and the Hawkes model and gives a good description of the related literature.

Reproducibility: Yes

Additional Feedback:


Review 2

Summary and Contributions: This paper proposes a model for continuous-time networks. Discrete-time networks allow for changes at arbitrary times whereas continuous-time networks do not have to be smooth. CHIP models timestamped relational data and is inspired by Block Hawkes Model (BHM). In BHM, the multiple edges between any pairs are dependent which adds more complexity to the model. CHIP on the other hand, considered such interactions to be independent. Under this model, the network can grow over time, the number of nodes can increase and more interaction can be added to the network. However, the number of clusters is fixed which prevents cluster changes in the model. The estimation procedure makes the paper reasonably well-motivated. The model is tested on a number of real-world datasets and I think the empirical results are impressive.

Strengths: Timestamped networks are getting more attention and research in this area is very interesting. The paper is overall well written and clear in its objectives and contributions. The experimental results are explained in detail and tested on relatively large datasets, which shows the scalability of the model. The provided code was modular and easy to follow. However, more comments on the code would be appreciated.

Weaknesses: The CHIP model is considering the number of clusters, or blocks to be fixed. The main weakness of SBMs is that their model can not be used for dynamic networks. In continuous-time networks, events between nodes can change the cluster behavior of those nodes, and therefore, the clusters are not fixed. This model can not be used for time-evolving networks. The generative process in section 3, should have been in a more mathematical format. The row concatenation can be replaced with “||”. The bold 1, is not defined, and therefore Y = Row concatenate(.) is not clear. Section 3.1 begins with the MLE for community assignments, however, there is no reference about it in the paper so far. Regarding citations, I suggest looking at a broader range of models on timestamped networks, not just the Poisson process or Hawkes process models. Any Bayesian model specially nonparametric versions on temporal networks can fit in this category. Although the experiments are compared to related work on this topic, I encourage the authors to pick a recent useful model on other stochastic processes and compare their model with them.

Correctness: This sentence in the introduction is not valid:”In the BHM, events between different pairs of nodes belonging to the same pair of communities are dependent, which makes it difficult to analyze. “ since in the cited paper they have are deviating from independence by the number of clusters as they mentioned:
The Asymptotic Independence Theorem demonstrates that pairs of adjacency matrix entries in the same block pair are dependent, but that the dependence is upper bounded by (3), and that the dependence goes to 0 for growing blocks. Grammar/Typos Introduction: - Paragraph 1: .. which consists of a set of objects and relationships between the objects. Related work: - Paragraph 3: or variational inference have “been” used in prior work Section 4.1 - Paragraph 2: There’s no need for the parenthesis.

Clarity: The paper is in general well written and the authors clearly communicate their objectives.

Relation to Prior Work: In related work, there’s a line referring to static networks that model diffusion from information cascades. It’s not clear enough what category these models fit in. Maybe classify network models according to full observations or partial observations and then mention static and dynamic networks from the information cascades that can be subsumed under partial observations. In the past two years there are models that have worked on discrete-time networks that are not cited here. The discrete-time counterparts referred in the introduction are a bit old, up to 2017. I think it’s worth mentioning more recent works in this area.

Reproducibility: Yes

Additional Feedback: in the experiment over mean test log-likelihood shows that for Enron best k is 2 and for Reality it’s 1. When the model clusters everything in 1 or 2, that means the clustering has not been well trained. The paper makes a reason that the log-likelihood is high because all pairs in a cluster are independent, how is that implies this? Could you please elaborate more on these results? What do you mean by any type of timestamped relational event data? How many types are there? and what feature of your model captures that? ---- Update ---- I appreciate the authors' response and thank their clarification on my points and other reviewer's points. The number of communities is still a limitation of the model and I encourage the authors to make the model more flexible in the future.


Review 3

Summary and Contributions: The paper describes a model, CHIP, of relational event data that integrates block community structure to a Hawkes Process, with independent processes at pairs. The model takes the latent/community structure of the Dubois-Smyth work, which is based on Poisson processes, and brings it up to speed with the work that’s been done on modeling social data using Hawkes Processes. The pairwise independence is somewhat simplistic, but dependence is tricky, and CHIP is able to fit/predict real data much better than the BHM model (WWW2019) that features dependence. The work presents nice theoretical results on the vanishing rate of msiclassication (Thm 1), and convergence/MSE of estimators (Thm 2, 3). The empirical work is a well-done synthetic data analysis as well as real-world data analysis, where the model does better than any existing methods. The work is clean and feels like the “right” model for the problem, to the point that it’s maybe surprising that nobody has added latent classes to a Hawkes process before. Which is usually a good sign. Update: I thank the authors for their responses.

Strengths: 1) Builds on both Hawkes process literature and latent-class Poisson process literature, combining them with rigorous analysis. 2) Well done empirics.

Weaknesses: 1) The work would be strengthened if there was a stronger case made for the utility of a good predictive model. Can this model be used for link prediction? Not really, because of the dyadic independence. Can it be used to populate type-ahead rankings in social products? Probably.

Correctness: The results appear to be correct.

Clarity: Yes.

Relation to Prior Work: Yes.

Reproducibility: Yes

Additional Feedback: - For an analog of the distinction between dependence vs. pairwise independence, see (Shalizi-Rinaldo, 2013) in the context of ERGMs and the key “dyadic independence assumption” there. - It would be helpful to better explain how the BHM model can't "revert" to the CHIP model by relaxing some sort of parameterization of the dependence. I was surprised to see such a difference in logL between BHM and CHIP even when k=1. - The title could be improved by making more clear that the work is Hawkes-process-based, and independent at that.


Review 4

Summary and Contributions: The authors of this work propose a novel approach to estimate a generative model of timestamped relational data based on the continuous-time Community Hawkes Independent Pairs (CHIP) process. The model can be seen as a generalization of the stochastic block model for timestamped events, where the timestamped edges between nodes are used to estimate the parameters of independent and univariate Hawkes processes for all pairs of $k$ communities. The authors combine this estimation procedure with a spectral clustering algorithm to assign nodes to a given number of $k$ communities, and use that assignment to estimate the model parameters. They show that, in the limit of an infinite number of nodes and an infinite observation time, the methods provides a consistent detection of communities. They further show that the parameter estimation is consistent and derive expressions for the distribution of their estimates, which can be used to obtain confidence intervals. Experiments in synthetic and empirical data on three social networks show that the method favorably compared to existing continuous-time generalizations of the stochastic block model.

Strengths: - The unsupervised learning problem addressed by this work, i.e. community detection in networks, is highly relevant for the NeurIPS community. The contribution is, to the best of my knowledge, novel and considerably advances the state of research in community detection in relational time series. In my view, key novelty of this work is neither the use of the Hawkes process for relational event modelling, nor the application of spectral clustering or the use of the stochastic block model to community detection, but rather the combination of those three approaches. - The authors thoroughly introduce the theoretical foundations of their work and analytically derive (i) the error rate of their method (even though for extremely simplified assumptions) and (ii) the asymptotic distributions for their estimator. These asymptotic distributions can be used to obtain confidence intervals for the estimates as well as an upper bound for the mean squared error, which is an interesting feature of the proposed method. The claims are sound and proofs to theorems are included in an extensive supplementary document. The empirical evaluation is reasonable and sufficiently backs up the claims of the authors. In summary this is an interesting and well-written paper that advances the state of research in graph clustering and relational event modelling.

Weaknesses: - I could not follow the motivation for the simplified estimation procedure (and subsequent theoretical analysis in 3.1) that ignores timestamps (using event counts to estimate the branching ratio). It would be helpful if the authors could explain what advantages this approach brings compared to a "standard" block model inference in networks with a weighted adjacency matrix and the assumption that "event counts" (i.e. edge weights) are generated by independent Bernoulli processes. For instance, how do the results for the misclustering error rate compare to existing works on dectectability limits in the (weighted) block model? I would guess that there is a difference due to the use of the variance of event counts within blocks, which is not used for parameter estimation in some of the simpler block modelling approaches. However, this simpler version of the model is not included in the experimental evaluation. In the authors' response, the motivation is clarified, and I think this additional discussion should be added in the camera-ready version. - An important challenge in community detection in graphs is to prevent overfitting in cases where the number of communities that we want to detect is not known a priori. Without additional precautions like, e.g. minimum description length or cross-validation techniques, a stochastic block model with a larger number of blocks trivially yield larger likelihoods, which renders a naive likelihood maximisation infeasible to determine the optimal number of blocks. I did not see this issue being discussed in the paper and I initially understood that this work mitigates this problem by means of spectral clustering (which could, in principle, also be used to determine the optimal number of blocks). However, algorithm 1 lists the number of blocks $k$ as an argument of the proposed method. For the synthetic networks generated from the CHIP model (in section 5.1 and 5.2) the number of blocks $k$ was fixed for any given experiment. For the comparison in real networks, the test log-likelihood is given for different values of $k$, which can effectively be viewed as a cross-validation to determine the optimal number of communities. I wonder about the scalability of this approach sine the parameter estimates have to be repeated for a potentially large number of values of $k$. The authors state that a fit for a single value of $k$ took less than a second on average but it is, e.g., unclear how that time depends on th value of $k$. Also, it would be great if the authors can discuss whether there may be better approaches to determine the optimal number of communities. I found the authors' response to this concern satisfactory, and I believe this could be addressed in the camera-ready version along the lines mentioned in the response. - It would be interesting to evaluate the method in data that includes ground truth community labels. Interestingly, such information is available for the RealityMining data set used by the authors, which has been collected via a proximity-sensing experiment that was accompanied by a survey (see e.g. ground truth on research group memberships in MIT campus data). Also, there are a number of other data sets where the performance of the method could be tested against a ground-truth, e.g. some of the SocioPatterns timestamped contact networks where ground truth information on groups (e.g. highschool/primary school classes, Facebook graphs, etc.) are available. I understand that the ground truth group structure may not have a 1:1 relationship to the definition of communities detected by the proposed method, i.e. we cannot expect that the method yields exactly the ground truth clusters. But it would be interesting to compare the performance of the proposed method with the competing methods in such a data set. I did not see a comment on this in the authors' response.

Correctness: To the best of my knowledge, the claims in the paper are correct and the empirical methodology is sound. The supplementary material contains well-documented code to reproduce the results and the three empirical data sets are included as well.

Clarity: The paper is very well-written. The authors do a great job explaining the theoretical basis of the work. This work is generally a pleasure to read.

Relation to Prior Work: The discussion how this work differs from prior works using continuous-time generalizations of the stochastic block model, and what advantages it provides, is rather short and could be extended. I understand that the authors may have struggled with the page limitations, since part of the discussion of related work has been moved to the supplement.

Reproducibility: Yes

Additional Feedback: Some additional questions and suggestions: - It would be great if the authors could clarify the notion of communities that they aim to detect with their method, specifically how the temporal patterns in the timestamped event sequence affect those communities. For instance, different from other works in the area of temporal community detection, the proposed method does not consider (i) a dynamic assignment of nodes to communities that cab change over time, (ii) a (static) assignment of nodes to communities is found that accounts for the temporal ordering of edges, or (iii) an assignment of nodes to communities such that the communities are maximally stable (also sometimes called "consistent") over time. Apart from utilizing the timestamps of events for the estimation of the parameters of the underlying model, if I understand correctly the temporal sequence of edges does not really affect the detected community structures. I think this should be clarified and the authors' response did not really shed new light on this critical part of the motivation. - In their initial treatment of the parameter estimation based on event counts in section 3.1, an estimation of the ratio $\alpha/\beta$ is given based on the mean and the variance of the event counts. Am I right to assume that this corresponds to an estimate of the branching ratio of the Hawkes process? If this is correct, this interpretation of the value in the context of the Hawkes process could be made clearer, possibly by referring to a brief discussion in the supplement. - I found the distinction between discrete- vs. continuous time models in the introduction confusing, as it appears to be the case that with "discrete-time" network models the authors actually refer to methods operating on static, time-aggregated snapshots (with no use of time whatsoever). This interpretation is exacerbated by the references given in that sentence. Maybe the authors could clarify what they mean with discrete time models? - It would be good if the authors could explicitly refer to the additional discussions or derivations in the supplementary material, where relevant in the main text. I found it difficult to understand which parts of the analysis or derivations have actually been moved to the supplement. - While reviewing this work, I came across two typos: line 99: have _been_ used in prior work line 164: are _the_ same for both set_s_ of parameters In light of the authors' response and the other reviewer comments, I decided to slightly reduce my score but I still keep up my positive assessment of this work.

[Author Response · NeurIPS 2020]

We thank the reviewers for their thoughtful feedback that shows they understood the key points in our paper. We
are glad that they found our contributions to be timely (R1, R2), relevant (R1, R4), novel (R4), and our paper to be
well-written (R1, R2, R4). We are particularly encouraged that they found both the theoretical rigor (R1, R3) and
empirical results (R2, R3) to be strong. An area of concern relates to the number of communities $k$ (R1, R2, R4)—we
first address this concern then respond to some other reviewer comments below.

**@R1 - The estimation of $k$ is not handled; @R4 - Are there better approaches**
**to find the optimal $k$?** We thank the reviewers for pointing this out. Our estimator
uses spectral clustering on the weighted adjacency matrix $N$ so model selection
approaches for static block models can be used. We used the eigengap heuristic for
the exploratory analysis in Section 5.4 and in C.2.3 and C.2.4 of the supplementary,
but more sophisticated methods including using eigenvalues of the non-backtracking
matrix and Bethe hessian matrix (Le & Levina, 2015), and network cross validation
(Chen & Lei, 2018; Li, Levina, & Zhu, 2020) could be used. Another approach
mentioned by R4 specific to timestamped networks, is to *hold out a portion of the*
*events and select the $k$ that maximizes test log-likelihood*, which we used in Table 1.
As shown in Figure 1, for $k < 100$, there is hardly any increase in the runtime, and
it is manageable even for $k = 1,000$. We would add this discussion to the paper.

Figure 1: CHIP's fitting runtime on the Facebook data on a log-log scale with increasing $k$.

**@R2, R4 - The number of communities $k$ and community assignments are fixed over time which prevents the**
**model to be used for dynamic network analysis:** Both $k$ and community assignments are indeed fixed in the CHIP
model and in most other continuous-time block models [1, 3-5, 7, 8]. This is a current limitation of continuous-time
block models compared to discrete-time models that often allow changes in communities over time [9-13]. However,
we disagree that this prevents the model from being used for dynamic network analysis because the temporal dynamics
are being captured by the Hawkes processes. *Thus, the CHIP model still captures time-varying behavior due to their*
*self-exciting nature despite the fixed communities.* Since the paper submission, we became aware of the continuous-time
block model of Corneli, Latouche, & Rossi (2018) that divides time into $D$ equally-spaced change points where
community structure can change. Such an approach could be used also with the CHIP model.

**@R1 - No detail on the likelihood estimation scheme proposed for $\alpha$ and $\beta$ and their theoretical properties:** The
estimation procedure is discussed in detail in Section A.3 in the supplementary. We have no guarantees for $\alpha$ and $\beta$ but
demonstrate in Section 5.2 through simulation that the MSEs of their estimators with decrease quadratically with $n$.

**@R1 - The paper concentrates on dense graphs. The dependence of parameter $\mu$ of Hawkes process on the node**
**size $n$ is not discussed in detail:** We provide results for the sparse regime in Section B.1.1 in the supplementary. We
let $\mu \asymp \frac{1}{f(n)g(T)}$, a function of $n$ and $T$ and explore various sparsity settings by varying $f$ and $g$. Our proofs allow
$\mu$ to vary with $n$ and $T$ and can be as small as $\log(n)/(nT)$, as R1 suggested. In particular, in the last paragraph we
wrote, "if we set $g(T) \asymp T$ and $f(n) = \frac{n}{\log n}$, such that $\mu_1 \asymp \mu_2 \asymp \frac{\log n}{nT}$, then the expected number of events between
a node pair is $O(\frac{\log n}{n})$. In that case, $r(T) \lesssim \frac{k^2}{\log n (c_1 - c_2)^2}$, and consistent community detection is possible as long as
$k = o(\sqrt{\log n}|c_1 - c_2|)$." We will add a discussion on the sparse graph setting and a reference to the supplementary.

**@R2 - Why does finding only 1 or 2 clusters suggests independence?** In CHIP, a small number of communities (e.g.
1 in the case of Reality Mining data) suggests a weak community structure, but not necessarily independence. That
conclusion was mostly derived from the fact that BHM (which models dependence of node pairs within block pairs)
achieves its best test log-likelihood on the same dataset for extremely large $k = 50$ on a network with only 70 nodes!

**@R3 - Is there a stronger case made for the utility of a good predictive model (in CHIP)?** We thank R3 for this
suggestion. Two potential use cases are for time-to-event prediction, i.e. the time until the next event between a pair of
nodes, and predicting the number of events between a pair of nodes in a future time period.

**@R3 - Why can't BHM turn into CHIP by a simple modification? Why such a high difference in log-likelihood**
**even when $k = 1$?** The BHM uses a single Hawkes process for each block pair then randomly assigns events to node
pairs so that the dependence between node pairs cannot be relaxed. On the other hand, CHIP assumes independent node
pairs in a block pair that share the same parameters. The closest the BHM can get to CHIP is for $k = 1$, where the
BHM shares parameters but has dependence, and for $k = n$, the BHM has independence but no parameter sharing.

**@R4 - What is the motivation for the simplified estimation procedure that ignores timestamps?** The main
advantage of ignoring timestamps is scalability—our estimators for the $\mu$ and $m$ parameters scale independent of the
number of events (beyond the trivial computation of the count matrix $N$), while the standard MLE using the timestamps
(e.g. in the BHM) requires solving a non-convex optimization problem that depends on the number of events.

We especially thank R4 for the very detailed comments and will incorporate them despite lack of space to respond here.

[Meta-Review · NeurIPS 2020]

The reviews are positive: The paper addresses an interesting problem, proposes a reasonably novel model, is well-executed, and makes an entertaining read. Although the reviewers point out a number of concerns (e.g. the dyadic independence assumption that will generally be unrealistic), the opinions and scores support publication. To my mind, the main weakness is perhaps that combining network models and Hawkes processes has been fashionable for a few years now, and although the model is technically novel, I have not found anything in the paper that I would consider unexpected.